# Maximum Entropy RL (Provably) Solves Some Robust RL Problems

**Benjamin Eysenbach**
Carnegie Mellon University, Google Brain
beysenba@cs.cmu.edu

**Sergey Levine**
UC Berkeley, Google Brain

## Abstract

Many potential applications of reinforcement learning (RL) require guarantees that the agent will perform well in the face of disturbances to the dynamics or reward function. In this paper, we prove theoretically that maximum entropy (MaxEnt) RL maximizes a lower bound on a robust RL objective, and thus can be used to learn policies that are robust to some disturbances in the dynamics and the reward function. While this capability of MaxEnt RL has been observed empirically in prior work, to the best of our knowledge our work provides the first rigorous proof and theoretical characterization of the MaxEnt RL robust set. While a number of prior robust RL algorithms have been designed to handle similar disturbances to the reward function or dynamics, these methods typically require additional moving parts and hyperparameters on top of a base RL algorithm. In contrast, our results suggest that MaxEnt RL by itself is robust to certain disturbances, without requiring any additional modifications. While this does not imply that MaxEnt RL is the best available robust RL method, MaxEnt RL is a simple robust RL method with appealing formal guarantees.

## 1 Introduction

Many real-world applications of reinforcement learning (RL) require control policies that not only maximize reward but continue to do so when faced with environmental disturbances, modeling errors, or errors in reward specification. These disturbances can arise from human biases and modeling errors, from non-stationary aspects of the environment, or actual adversaries acting in the environment. A number of works have studied how to train RL algorithms to be robust to disturbances in the environment (e.g., Morimoto & Doya (2005); Pinto et al. (2017); Tessler et al. (2019)). However, designing robust RL algorithms requires care, typically requiring an adversarial optimization problem and introducing additional hyperparameters (Pinto et al., 2017; Tessler et al., 2019). Instead of designing a new robust RL algorithm, we will instead analyze whether an existing RL method, MaxEnt

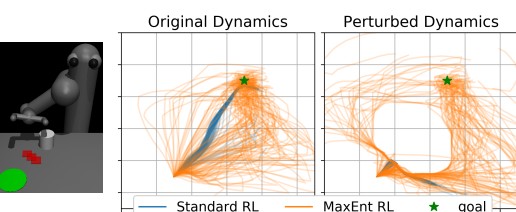

Figure 1: **MaxEnt RL is robust to disturbances.** *(Left)* We applied both standard RL and MaxEnt RL to a manipulation task without obstacles, but added obstacles (red squares) during evaluation. We then plot the position of the object when evaluating the learned policies *(Center)* on the original environment and *(Right)* on the new environment with an obstacle. The stochastic policy learned by MaxEnt RL often navigates around the obstacle, whereas the deterministic policy from standard RL almost always collides with the obstacle.

RL, offers robustness to perturbations. MaxEnt RL methods are based on the maximum entropy principle, and augmented the expected reward objective with an entropy maximization term (Ziebart et al., 2008). Prior work has conjectured that such algorithms should learn robust policies (Haarnoja et al., 2017; Huang et al., 2019), on account of the close connection between maximum entropy methods and robustness in supervised learning domains (Grünwald et al., 2004; Ziebart, 2010). However, despite decades of research on maximum entropy RL methods (Kappen, 2005; Todorov, 2007; Toussaint, 2009; Theodorou et al., 2010), a formal characterization of the robustness properties of such approaches has proven elusive. To our knowledge, no prior work formally proves that MaxEnt RL methods are robust, and no prior work characterizes the disturbances against which these methods

are robust. Showing how to obtain robust policies from existing MaxEnt RL methods, which already constitute a significant portion of the RL methods in use today (Abdolmaleki et al., 2018; Haarnoja et al., 2018a; Vieillard et al., 2020), would be useful because it would enable practitioners to leverage existing, tried-and-true methods to solve robust RL problems.

Stochastic policies, of the sort learned with MaxEnt RL, inject noise into the actions during training, thereby preparing themselves for deployment in environments with disturbances. For example, in the robot pushing task shown in Fig. 1, the policy learned by MaxEnt RL pushes the white puck to the goal using many possible routes. In contrast, (standard) RL learns a deterministic policy, always using the same route to get to the goal. Now, imagine that this environment is perturbed by adding the red barrier in Fig. 1. While the policy learned by (standard) RL always collides with this obstacle, the policy learned by MaxEnt RL uses many routes to solve the task, and some fraction of these routes continue to solve the task even when the obstacle is present. While a number of prior works have articulated the intuition that the stochastic policies learned via MaxEnt RL should be robust to disturbances (Levine, 2018; Abdolmaleki et al., 2018; Lee et al., 2019), no prior work has actually shown that MaxEnt RL policies are provably robust, nor characterized the *set* of disturbances to which they are robust. Applications of MaxEnt RL methods to problems that demand robustness are likely hampered by a lack of understanding of *when* such methods are robust, what kinds of reward functions should be used to obtain the desired type of robustness, and how the task should be set up. The goal in our work is to make this notion precise, proving that MaxEnt RL is already a robust RL algorithm, and deriving the robust set for these policies.

The main contribution of this paper is a theoretical characterization of the robustness of existing MaxEnt RL methods. Specifically, we show that the MaxEnt RL objective is a nontrivial *lower bound* on the robust RL objective, for a particular robust set. Importantly, these two objectives use different reward functions: to learn a policy that maximizes a reward function on a wide range of dynamics functions, one should apply MaxEnt RL to a *pessimistic* version of that reward function. MaxEnt RL is *not* robust with respect to the reward function it is trained on. As we will show, the robust set for these methods is non-trivial, and in some important special cases can be quite intuitive. To our knowledge, our results are the first to formally show robustness of standard MaxEnt RL methods to dynamics perturbations and characterize their robust set. Importantly, our analysis and experiments highlight that robustness is only achieved for relatively large entropy coefficients. We validate our theoretical results on a set of illustrative empirical experiments.

## 2   RELATED WORK

The study of robustness in controls has a long history, with robust or $H_\infty$ control methods proposing provably robust solutions under various assumptions on the true dynamics (Zhou et al., 1996; Doyle et al., 2013). RL offers an appealing alternative, since robust RL methods can in principle learn robust policies *without* knowing the true dynamics. However, policies learned by standard RL algorithms often flounder in the face of environmental disturbances (Rajeswaran et al., 2016; Peng et al., 2018). The problem of learning robust policies that achieve high reward in the face of *adversarial* environmental disturbances, often called *robust RL*, has been well studied in the literature (Bagnell et al., 2001; Nilim & Ghaoui, 2003; Morimoto & Doya, 2005; Pinto et al., 2017; Tessler et al., 2019; Russel & Petrik, 2019; Kamalaruban et al., 2020; Russel et al., 2020; 2021; Derman et al., 2021). Prior robust RL methods are widely-applicable but often require many additional hyperparameters or components. For example, Bagnell et al. (2001) modify the Bellman backup in a Q-learning by solving a convex optimization problem in an inner loop, and Tessler et al. (2019) trains an additional adversary policy via RL.

Robust RL is different from robust control in the traditional sense, which focuses on stability independent of any reward function (Zhou et al., 1996; Doyle et al., 2013). Robust RL problem involves estimating a policy's returns under "similar" MDPs, a problem that has been studied in terms of distance metrics for MDPs (Lecarpentier et al., 2020). Robust RL is different from maximizing the *average* reward across many environments, as done by methods such as domain randomization (Sadeghi & Levine, 2016; Rajeswaran et al., 2016; Peng et al., 2018). Closely related to robust RL are prior methods that are robust to disturbances in the reward function (Hadfield-Menell et al., 2017; Bobu et al., 2020; Michaud et al., 2020), or aim to minimize a cost function in addition to maximizing reward (Chow et al., 2017; Achiam et al., 2017; Chow et al., 2019; Carrara et al., 2019; Tang et al., 2020; Thananjeyan et al., 2021). Robust RL is also different from the problem of

learning transferable or generalizable RL agents (Lazaric, 2012; Zhang et al., 2018; Justesen et al., 2018; Cobbe et al., 2019), which focuses on the average-case performance on new environments, rather than the worst-case performance.

## 3 PRELIMINARIES

We assume that an agent observes states $\mathbf{s_t}$, takes actions $\mathbf{a_t} \sim \pi(\mathbf{a_t} \mid \mathbf{s_t})$, and obtains rewards $r(\mathbf{s_t}, \mathbf{a_t})$. The initial state is sampled $\mathbf{s_1} \sim p_1(\mathbf{s_1})$, and subsequent states are sampled $\mathbf{s_{t+1}} \sim p(\mathbf{s_{t+1}} \mid \mathbf{s_t}, \mathbf{a_t})$. We will use $p^\pi(\tau)$ to denote the probability distribution over trajectories for policy $\pi$. Episodes have $T$ steps, which we summarize as a trajectory $\tau \triangleq (\mathbf{s_1}, \mathbf{a_1}, \cdots, \mathbf{s_T}, \mathbf{a_T})$. Without loss of generality, we can assume that rewards are undiscounted, as any discount can be addressed by modifying the dynamics to transition to an absorbing state with probability $1 - \gamma$. The standard RL objective is:

$$\arg\max_\pi \mathbb{E}_{\tau \sim p^\pi(\tau)}\left[\sum_{t=1}^T r(\mathbf{s_t}, \mathbf{a_t})\right], \quad \text{where} \quad p^\pi(\tau) = p_1(\mathbf{s_1})\prod_{t=1}^T p(\mathbf{s_{t+1}} \mid \mathbf{s_t}, \mathbf{a_t})\pi(\mathbf{a_t} \mid \mathbf{s_t})d\tau$$

is the distribution over trajectories when using policy $\pi$. In fully observed MDPs, there always exists a deterministic policy as a solution (Puterman, 2014). The MaxEnt RL objective is to maximize the sum of expected reward and conditional action entropy. Formally, we define this objective in terms of a policy $\pi$, the dynamics $p$ and the reward function $r$:

$$J_{\text{MaxEnt}}(\pi; p, r) \triangleq \mathbb{E}_{\mathbf{a_t} \sim \pi(\mathbf{a_t}|\mathbf{s_t}), \mathbf{s_{t+1}} \sim p(\mathbf{s_{t+1}}|\mathbf{s_t}, \mathbf{a_t})}\left[\sum_{t=1}^T r(\mathbf{s_t}, \mathbf{a_t}) + \alpha\mathcal{H}_\pi[\mathbf{a_t} \mid \mathbf{s_t}]\right], \quad (1)$$

where $\mathcal{H}_\pi[\mathbf{a_t} \mid \mathbf{s_t}] = \int_\mathcal{A} \pi(\mathbf{a_t} \mid \mathbf{s_t})\log\frac{1}{\pi(\mathbf{a_t}|\mathbf{s_t})}d\mathbf{a_t}$ denotes the entropy of the action distribution. The *entropy coefficient* $\alpha$ balances the reward term and the entropy term; we use $\alpha = 1$ in our analysis. As noted above, the discount factor is omitted because it can be subsumed into the dynamics.

Our main result will be that maximizing the MaxEnt RL objective (Eq. 1) results in robust policies. We quantify robustness by measuring the reward of a policy when evaluated on a *new* reward function $\tilde{r}$ or dynamics function $\tilde{p}$, which is chosen adversarially from some set:

$$\max_\pi \min_{\tilde{p} \in \tilde{\mathcal{P}}, \tilde{r} \in \tilde{\mathcal{R}}} \mathbb{E}_{\tilde{p}(\mathbf{s_{t+1}}|\mathbf{s_t}, \mathbf{a_t}), \pi(\mathbf{a_t}|\mathbf{s_t})}\left[\sum_{t=1}^T \tilde{r}(\mathbf{s_t}, \mathbf{a_t})\right].$$

This *robust RL* objective is defined in terms of the sets of dynamics $\tilde{\mathcal{P}}$ and reward functions $\tilde{\mathcal{R}}$. Our goal is to characterize these sets. The robust RL objective can be interpreted as a two-player, zero-sum game. The aim is to find a Nash equilibrium. Our goal is to prove that MaxEnt RL (with a *different reward function*) optimizes a lower bound this robust objective, and to characterize the robust sets $\tilde{\mathcal{P}}$ and $\tilde{\mathcal{R}}$ for which this bound holds.

## 4 MAXENT RL AND ROBUST CONTROL

In this section, we prove the conjecture that MaxEnt RL is robust to disturbances in the environment. Our main result is that MaxEnt RL can be used to maximize a lower bound on a certain robust RL objective. Importantly, doing this requires that MaxEnt RL be applied to a different, pessimistic, version of the target reward function. Before presenting our main result, we prove that MaxEnt RL is robust against disturbances to the reward function. This result is a simple extension of prior work, and we will use this result for proving our main result about dynamics robustness

### 4.1 ROBUSTNESS TO ADVERSARIAL REWARD FUNCTIONS

We first show that MaxEnt RL is robust to some degree of misspecification in the reward function. This result may be useful in practical settings with learned reward functions (Fu et al., 2018; Xu & Denil, 2019; Michaud et al., 2020) or misspecified reward function (Amodei et al., 2016; Clark & Amodei, 2016). Precisely, the following result will show that applying MaxEnt RL to one reward function, $r(\mathbf{s_t}, \mathbf{a_t})$, results in a policy that is guaranteed to also achieve high return on a range of other reward functions, $\tilde{r}(\mathbf{s_t}, \mathbf{a_t}) \in \tilde{\mathcal{R}}$:

**Theorem 4.1.** *Let dynamics $p(\mathbf{s_{t+1}} \mid \mathbf{s_t}, \mathbf{a_t})$, policy $\pi(\mathbf{a_t} \mid \mathbf{s_t})$, and reward function $r(\mathbf{s_t}, \mathbf{a_t})$ be given. Assume that the reward function is finite and that the policy has support everywhere (i.e., $\pi(\mathbf{a_t} \mid \mathbf{s_t}) > 0$ for all states and actions). Then there exists a positive constant $\epsilon > 0$ such that the MaxEnt RL objective $J_{MaxEnt}$ is equivalent to the robust RL objective defined by the robust set $\tilde{\mathcal{R}}(\pi)$:*

$$\min_{\tilde{r} \in \tilde{\mathcal{R}}(\pi)} \mathbb{E}\left[\sum_t \tilde{r}(\mathbf{s_t}, \mathbf{a_t})\right] = J_{MaxEnt}(\pi; p, r) \quad \forall \pi,$$

*where the adversary chooses a reward function from the set*

$$\tilde{\mathcal{R}}(\pi) \triangleq \left\{ \tilde{r}(\mathbf{s_t}, \mathbf{a_t}) \; \middle| \; \mathbb{E}_\pi\left[\sum_t \log \int_{\mathcal{A}} \exp(r(\mathbf{s_t}, \mathbf{a_t}') - \tilde{r}(\mathbf{s_t}, \mathbf{a_t}'))d\mathbf{a_t}'\right] \leq \epsilon \right\}. \tag{2}$$

Thus, when we use MaxEnt RL with some reward function $r$, the policy obtained is guaranteed to also obtain high reward on all similar reward functions $\tilde{r}$ that satisfy Eq. 2. We discuss how MaxEnt RL can be used to learn policies robust to arbitrary sets of reward functions in Appendix A.10. The proof can be found in Appendix A.2, and is a simple extension of prior work (Grünwald et al., 2004; Ziebart et al., 2011). Our proof does not assume that the reward function is bounded nor that the policy is convex. While the proof is straightforward, it will be useful as an intermediate step when proving robustness to dynamics in the next subsection.

The robust set $\tilde{\mathcal{R}}$ corresponds to reward functions $\tilde{r}$ that are not too much smaller than the original reward function: the adversary cannot decrease the reward for any state or action too much. The robust set $\tilde{\mathcal{R}}$ depends on the policy, so the adversary has the capability of looking at which states the policy visits before choosing the adversarial reward function $\tilde{r}$. For example, the adversary may choose to apply larger perturbations at states and actions that the agent frequently visits.

## 4.2 ROBUSTNESS TO ADVERSARIAL DYNAMICS

We now show that MaxEnt RL learns policies that are robust to perturbations to the dynamics. Importantly, to have MaxEnt RL learn policies that robustly maximize one a reward function, we will apply MaxEnt RL to a different, pessimistic reward function:

$$\bar{r}(\mathbf{s_t}, \mathbf{a_t}, \mathbf{s_{t+1}}) \triangleq \frac{1}{T} \log r(\mathbf{s_t}, \mathbf{a_t}) + \mathcal{H}[\mathbf{s_{t+1}} \mid \mathbf{s_t}, \mathbf{a_t}]. \tag{3}$$

The $\log(\cdot)$ transformation is common in prior work on learning risk-averse policies (Mihatsch & Neuneier, 2002). The entropy term rewards the policy for vising states that have stochastic dynamics, which should make the policy harder for the adversary to exploit. In environments that have the same stochasticity at every state (e.g., LG dynamics), this entropy term becomes a constant and can be ignored. In more general settings, computing this pessimistic reward function would require some knowledge of the dynamics. Despite this limitation, we believe that our results may be of theoretical interest, taking a step towards explaining the empirically-observed robustness properties of MaxEnt RL.

To formally state our main result, we must define the range of "similar" dynamics functions against which the policy will be robust. We use the following divergence between two dynamics functions:

$$d(p, \tilde{p}; \tau) \triangleq \sum_{\mathbf{s_t} \in \tau} \log \iint_{\mathcal{A} \times \mathcal{S}} \frac{p(\mathbf{s_{t+1}}' \mid \mathbf{s_t}, \mathbf{a_t}')}{\tilde{p}(\mathbf{s_{t+1}}' \mid \mathbf{s_t}, \mathbf{a_t}')} d\mathbf{a_t}' d\mathbf{s_{t+1}}'. \tag{4}$$

This divergence is large when the adversary's dynamics $\tilde{p}(\mathbf{s_{t+1}} \mid \mathbf{s_t}, \mathbf{a_t})$ assign low probability to a transition with high probability in the training environment $\tilde{p}$. Our main result shows that applying MaxEnt RL to the pessimistic reward function results in a policy that is robust to these similar dynamics functions:

**Theorem 4.2.** *Let an MDP with dynamics $p(\mathbf{s_{t+1}} \mid \mathbf{s_t}, \mathbf{a_t})$ and reward function $r(\mathbf{s_t}, \mathbf{a_t}) > 0$ be given. Assume that the dynamics have finite entropy (i.e., $\mathcal{H}[\mathbf{s_{t+1}} \mid \mathbf{s_t}, \mathbf{a_t}]$ is finite for all states and actions). Then there exists a constant $\epsilon > \mathcal{H}_{\pi^*}[\mathbf{a_t} \mid \mathbf{s_t}]$ such that the MaxEnt RL objective with dynamics $p$ and reward function $\bar{r}$ is a lower bound on the robust RL objective*

$$\min_{\tilde{p} \in \tilde{\mathcal{P}}(\pi)} J_{MaxEnt}(\pi; \tilde{p}, r) \geq \exp(J_{MaxEnt}(\pi; p, \bar{r}) + \log T),$$

*where the robust set is defined as*

$$\tilde{\mathcal{P}} \triangleq \left\{ \tilde{p}(\mathbf{s}' \mid \mathbf{s}, \mathbf{a}) \,\middle|\, \mathbb{E}_{\substack{\pi(\mathbf{a_t}|\mathbf{s_t}) \\ p(\mathbf{s_{t+1}}|\mathbf{s_t}, \mathbf{a_t})}} [d(p, \tilde{p}; \tau)] \le \epsilon \right\}. \tag{5}$$

In defining the robust set, the adversary chooses a dynamics function from the robust set $\tilde{\mathcal{P}}$ independently at each time step; our next result (Lemma 4.3) will describe the value of $\epsilon$ in more detail. The proof, presented in Appendix A.4, first shows the robustness to reward perturbations implies robustness to dynamics perturbations and then invokes Theorem 4.1 to show that MaxEnt RL learns policies that are robust to reward perturbations.

This result can be interpreted in two ways. First, if a user wants to acquire a policy that optimizes a reward under many possible dynamics, we should run MaxEnt RL with a specific, pessimistic reward function $\bar{r}(\mathbf{s_t}, \mathbf{a_t}, \mathbf{s_{t+1}})$. This pessimistic reward function depends on the environment dynamics, so it may be hard to compute without prior knowledge or a good model of the dynamics. However, in some settings (e.g., dynamics with constant additive noise), we might assume that $\mathcal{H}[\mathbf{s_{t+1}} \mid \mathbf{s_t}, \mathbf{a_t}]$ is approximately constant, in which case we simply set the MaxEnt RL reward to $\bar{r}(\mathbf{s_t}, \mathbf{a_t}, \mathbf{s_{t+1}}) = \log r(\mathbf{s_t}, \mathbf{a_t})$. Second, this result says that every time a user applies MaxEnt RL, they are (implicitly) solving a robust RL problem, one defined in terms of a different reward function. This connection may help explain why prior work has found that the policies learned by MaxEnt RL tend to be robust against disturbances to the environment (Haarnoja et al., 2019).

Theorem 4.2 relates one MaxEnt RL problem to another (robust) MaxEnt RL problem. We can also show that MaxEnt RL maximizes a lower bound on an *unregularized* RL problem.

**Corollary 4.2.1.** *Under the same assumptions as Theorem 4.1 and 4.2, the MaxEnt RL problem is a lower bound on the robust RL objective:*

$$\min_{\tilde{p} \in \tilde{\mathcal{P}}(\pi), \tilde{r} \in \tilde{\mathcal{R}}(\pi)} \mathbb{E}_{\tilde{p}(\mathbf{s_{t+1}}|\mathbf{s_t}, \mathbf{a_t}), \pi(\mathbf{a_t}|\mathbf{s_t})} \left[ \sum_t \tilde{r}(\mathbf{s_t}, \mathbf{a_t}) \right] \ge \exp(J_{MaxEnt}(\pi; p, \bar{r}) + \log T),$$

*where the robust sets $\tilde{\mathcal{P}}(\pi)$ and $\tilde{\mathcal{R}}(\pi)$ as defined as in Theorem 4.1 and 4.2.*

Our next result analyzes the size of the robust set by providing a lower bound on $\epsilon$. Doing so proves that Theorem 4.2 is non-vacuous and will also tell us how to increase the size of the robust set.

**Lemma 4.3.** *Assume that the action space is discrete. Let a reward function $r(\mathbf{s_t}, \mathbf{a_t})$ and dynamics $p(\mathbf{s_{t+1}} \mid \mathbf{s_t}, \mathbf{a_t})$ be given. Let $\pi(\mathbf{a_t} \mid \mathbf{s_t})$ be the corresponding policy learned by MaxEnt RL. Then the size of $\epsilon$ (in Eq. 5) satisfies the following:*

$$\epsilon = T \cdot \mathbb{E}_{\substack{\mathbf{a_t} \sim \pi(\mathbf{a_t}|\mathbf{s_t}), \\ \mathbf{s_{t+1}} \sim p(\mathbf{s_{t+1}}|\mathbf{s_t}, \mathbf{a_t})}} [\mathcal{H}_{\tilde{p}}[\mathbf{s_{t+1}} \mid \mathbf{s_t}, \mathbf{a_t}] + \mathcal{H}_\pi[\mathbf{a_t} \mid \mathbf{s_t}]] \ge T \cdot \mathbb{E}_{\substack{\mathbf{a_t} \sim \pi(\mathbf{a_t}|\mathbf{s_t}), \\ \mathbf{s_{t+1}} \sim p(\mathbf{s_{t+1}}|\mathbf{s_t}, \mathbf{a_t})}} [\mathcal{H}_\pi[\mathbf{a_t} \mid \mathbf{s_t}]].$$

This result provides an exact expression for $\epsilon$. Perhaps more interesting is the inequality, which says that the size of the robust set is at least as large as the policy's entropy. For example, if MaxEnt RL learns a policy with an entropy of 10 bits, then $\epsilon \ge 10$. This result immediately tells us how to increase the size of the robust set: run MaxEnt RL with a larger entropy coefficient $\alpha$. Many popular implementations of MaxEnt RL automatically tune the entropy coefficient $\alpha$ so that the policy satisfies an entropy constraint (Haarnoja et al., 2018b). Our derivation here suggests that the entropy constraint on the policy corresponds to a lower bound on the size of the robust set.

The constraint in the definition of the robust set holds in expectation. Thus, the adversary can make large perturbations to some transitions and smaller perturbations to other states, as long as the average perturbation is not too large. Intuitively, the constraint value $\epsilon$ is the adversary's "budget," which it can use to make large changes to the dynamics in just a few states, or to make smaller changes to the dynamics in many states.

## 4.3 WORKED EXAMPLES

This section provides worked examples of our robustness results. The aim is to build intuition for what our results show and to show that, in simple problems, the robust set has an intuitive interpretation.

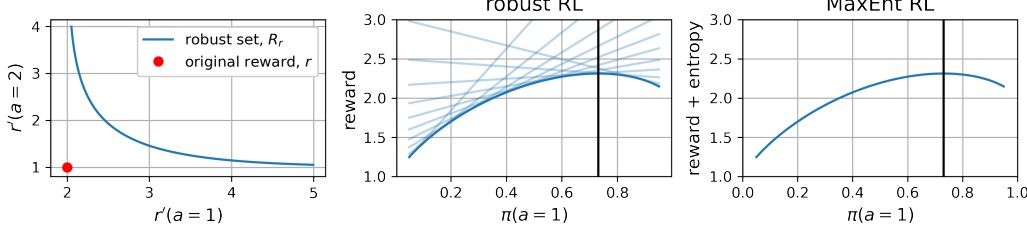

Figure 2: **MaxEnt RL and Robustness to Adversarial Reward Functions**: *(Left)* Applying MaxEnt RL to one reward function (red dot) yields a policy that is guaranteed to get high reward on many other reward functions (blue curve). *(Center)* For each reward function $(r(a = 1), r(a = 2))$ on that blue curve, we evaluate the expected return of a stochastic policy. The robust RL problem (for rewards) is to choose the policy whose worst-case reward (dark blue line) is largest. *(Right)* Plotting the MaxEnt RL objective (Eq. 1) for those same policies, we observe that the MaxEnt RL objective is identical to the robust RL objective .

**Reward robustness.** We present two worked examples of reward robustness. To simplify analysis, we consider the following, smaller robust set, which effectively corresponds to a weaker adversary:

$$\left\{ r(\mathbf{s_t}, \mathbf{a_t}) \,\middle|\, \log \int_{\mathcal{A}} \exp(r(\mathbf{s_t}, \mathbf{a_t}) - \tilde{r}(\mathbf{s_t}, \mathbf{a_t})) d\mathbf{a_t}' \leq \frac{\epsilon}{T} \right\} \subseteq \tilde{\mathcal{R}}. \tag{6}$$

For the **first example**, define a 2-armed bandit with the following reward function and corresponding robust set:

$$r(\mathbf{a}) = \begin{cases} 2 & \mathbf{a} = 1 \\ 1 & \mathbf{a} = 2 \end{cases}, \quad \tilde{\mathcal{R}} = \left\{ \tilde{r} \,\middle|\, \log \int_{\mathcal{A}} \exp(r(\mathbf{a}) - \tilde{r}(\mathbf{a})) d\mathbf{a} \leq \epsilon \right\}.$$

Fig. 2 (left) plots the original reward function, $r$ as a red dot, and the collection of reward functions, $\tilde{\mathcal{R}}$, as a blue line. In the center plot we plot the expected reward for each reward in $\tilde{\mathcal{R}}$ as a function of the action $\mathbf{a}$. The robust RL problem in this setting is to choose the policy whose worst-case reward (dark blue line) is largest. The right plot shows the MaxEnt RL objective. We observe that the robust RL objective and the MaxEnt RL objectives are equivalent.

For the **second example**, we use a task with a 1D, bounded action space $\mathcal{A} = [-10, 10]$ and a reward function composed of a task-specific reward $r_{\text{task}}$ and a penalty for deviating from some desired action $\mathbf{a}^*$: $r(\mathbf{s}, \mathbf{a}) \triangleq r_{\text{task}}(\mathbf{s}, \mathbf{a}) - (\mathbf{a} - \mathbf{a}^*)^2$. The adversary will perturb this desired action by an amount $\Delta a$ and decrease the weight on the control penalty by 50%, resulting in the following reward function: $\tilde{r}(\mathbf{s}, \mathbf{a}) \triangleq r_{\text{task}}(\mathbf{s}, \mathbf{a}) - \frac{1}{2}(\mathbf{a} - (\mathbf{a}^* + \Delta \mathbf{a}))^2$. In this example, the subset of the robust set in Eq. 6 corresponds to perturbations $\Delta \mathbf{a}$ that satisfy

$$\Delta \mathbf{a}^2 + \frac{1}{2} \log(2\pi) + \log(20) \leq \frac{\epsilon}{T}.$$

Thus, MaxEnt RL with reward function $r$ yields a policy that is robust against adversaries that perturb $\mathbf{a}^*$ by at most $\Delta \mathbf{a} = \mathcal{O}(\sqrt{\epsilon})$. See Appendix A.3 for the full derivation.

**Dynamics robustness.** The set of dynamics we are robust against, $\tilde{\mathcal{P}}$, has an intuitive interpretation as those that are sufficiently close to the original dynamics $p(\mathbf{s_{t+1}} \mid \mathbf{s_t}, \mathbf{a_t})$. This section shows that, in the case of linear-Gaussian dynamics (described at the end of this subsection), this set corresponds to a bounded L2 perturbation of the next state.

Because the robust set in Theorem 4.2 is defined in terms of the policy, the adversary can intelligently choose where to perturb the dynamics based on the policy's behavior. Robustness against this adversary also guarantees robustness against an adversary with a smaller robust set, that does not depend on the policy:

$$\left\{ \tilde{p}(\mathbf{s_{t+1}} \mid \mathbf{s_t}, \mathbf{a_t}) \,\middle|\, \log \iint_{\mathcal{A} \times \mathcal{S}} e^{\log p(\mathbf{s_{t+1}}|\mathbf{s_t},\mathbf{a_t}) - \log \tilde{p}(\mathbf{s_{t+1}}|\mathbf{s_t},\mathbf{a_t})} d\mathbf{a_t}' d\mathbf{s_{t+1}}' \leq \frac{\epsilon}{T} \right\} \subseteq \tilde{\mathcal{P}}.$$

We will use this subset of the robust set in the following worked example.

Consider an MDP with 1D, bounded, states and actions $\mathbf{s_t}, \mathbf{a_t} \in [-10, 10]$. The dynamics are $p(\mathbf{s_{t+1}} \mid \mathbf{s_t}, \mathbf{a_t}) = \mathcal{N}(\mathbf{s_{t+1}}; \mu = A\mathbf{s_t} + B\mathbf{a_t}, \sigma = 1)$ the reward function is $r(\mathbf{s_t}, \mathbf{a_t}) = \|\mathbf{s_t}\|_2^2$, and episodes have finite length $T$. Note that the dynamics entropy $\mathcal{H}[\mathbf{s_{t+1}} \mid \mathbf{s_t}, \mathbf{a_t}]$ is constant. We assume

Figure 3: MaxEnt RL is competitive with prior robust RL methods.

the adversary modifies the dynamics by increasing the standard deviation to $\sqrt{2}$ and shifts the bias by an amount $\beta$, resulting in the dynamics $\tilde{p}(\mathbf{s_{t+1}} \mid \mathbf{s_t}, \mathbf{a_t}) = \mathcal{N}(\mathbf{s_{t+1}}; \mu = A\mathbf{s_t} + B\mathbf{a_t} + \beta, \sigma = \sqrt{2})$. The robust set defined in Theorem 4.2 specifies that the adversary can choose any value of $\beta$ that satisfies $\frac{1}{2}\beta^2 + \log(8\sqrt{\pi}) + \log(20) \leq \epsilon$. To apply MaxEnt RL to learn a policy that is robust to any of these adversarial dynamics, we would use the pessimistic reward function specified by Theorem 4.2: $\bar{r}(\mathbf{s_t}, \mathbf{a_t}, \mathbf{s_{t+1}}) = \frac{2}{T}\log\|\mathbf{s_t}\|_2$. See Appendix A.7 for the full derivation.

### 4.4 LIMITATIONS OF ANALYSIS

We identify a few limitations of our analysis that may provide directions for future work. First, our definition of the robust set is different from the more standard $H_\infty$ and KL divergence constraint sets used in prior work. Determining a relationship between these two different sets would allow future work to claim that MaxEnt RL is also robust to these more standard constraint sets (see Lecarpentier et al. (2020)). On the other hand, MaxEnt RL may not be robust to these more conventional constraint sets. Showing this may inform practitioners about what sorts of robustness they should *not* except to reap from MaxEnt RL. A second limitation is the construction of the augmented reward: to learn a policy that will maximize reward function $r$ under a range of possible dynamics functions, our theory says to apply MaxEnt RL to a different reward function, $\bar{r}$, which includes a term depending on the dynamics entropy. While this term can be ignored in special MDPs that have the same stochasticity at every state, in more general MDPs it will be challenging to estimate this augmented reward function. Determining more tractable ways to estimate this augmented reward function is an important direction for future work.

### 5 NUMERICAL SIMULATIONS

This section will present numerical simulations verifying our theoretical result that MaxEnt RL is robust to disturbances in the reward function and dynamics function. We describe our precise implementation of MaxEnt RL, standard RL, and all environments in Appendix B.

**Comparison with prior robust RL methods.** We compare MaxEnt RL against two recent robust RL methods, PR-MDP and NR-MDP (Tessler et al., 2019). These methods construct a two-player game between a player that chooses actions and a player that perturbs the dynamics or actions. This recipe for robust RL is common in prior work (Pinto et al., 2017), and we choose to compare to Tessler et al. (2019) because it is a recent, high-performing instantiation of this recipe. We also include the DDPG baseline from Tessler et al. (2019). We evaluate all methods on the benchmark tasks used in Tessler et al. (2019), which involve evaluating in an environment where the masses are different from the training environment. We ran MaxEnt RL with both small and large entropy coefficients, evaluating the final policy for 30 episodes and taking the average performance. We repeated this for 5 random seeds to obtain the standard deviation. The results shown in Fig. 3 suggest that MaxEnt RL *with a large entropy coefficient $\alpha$ is at least competitive, if not better,* than prior purpose-designed robust RL methods. Note that MaxEnt RL is substantially simpler than PR-MDP and NR-MDP.

**Intuition for why MaxEnt RL is robust.** To build intuition into why MaxEnt RL should yield robust policies, we compared MaxEnt RL (SAC (Haarnoja et al., 2018a)) with standard RL (TD3 (Fujimoto et al., 2018)) on two tasks. On the pusher task, shown in Fig. 1, a new obstacle was added during evaluation. On the button task, shown in Fig. 4b, the box holding the button was moved closer or further away from the robot during evaluation. *Note that the robust RL objective corresponds to an adversary choosing the worst-case disturbances to these environments.*

Fig. 1 *(right)* and Fig. 4b *(center)* show that MaxEnt RL has learned many ways of solving these tasks, using different routes to push the puck to the goal and using different poses to press the button. Thus,

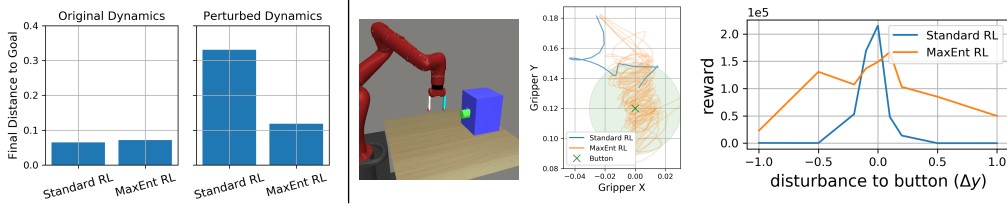

(a) Pusher: New Obstacle

(b) Button: Moved Goal

Figure 4: **Robustness to changes in the dynamics**: MaxEnt RL policies learn many ways of solving a task, making them robust to perturbations such as *(Left)* new obstacles and *(Right)* changes in the goal location.

when we evaluate the MaxEnt RL policy on perturbed environments, it is not surprising that some of these strategies continue to solve the task. In contrast, the policy learned by standard RL always uses the same strategy to solve these tasks, so the agent fails when a perturbation to the environment makes this strategy fail. Quantitatively, Fig. 4a and 4b (right) show that the MaxEnt RL policy is more robust than a policy trained with standard RL.

In many situations, simply adding noise to the deterministic policy found by standard RL can make that policy robust to some disturbances. MaxEnt RL does something more complex, dynamically adjusting the amount of noise depending on the current state. This capability allows MaxEnt RL policies to have lower entropy in some states as needed to ensure high reward. We study this capability in the 2D navigation task shown in Fig. 5. The agent starts near the top left corner and gets reward for navigating to the bottom right hand corner, but incurs a large cost for entering the red regions. The policy learned

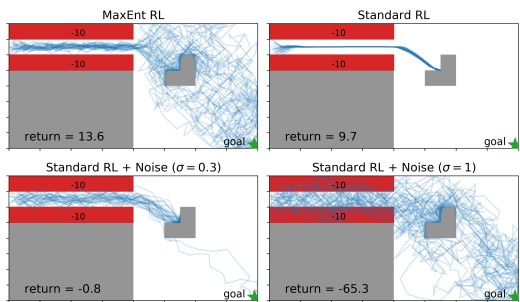

Figure 5: MaxEnt RL is not standard RL + noise.

by MaxEnt RL has low entropy initially to avoid colliding with the red obstacles, and then increases its entropy in the second half of the episode. To study robustness, we introduce a new "L"-shaped obstacle for evaluation. The policy learned by MaxEnt RL often navigates around this obstacle, whereas the policy from standard RL always collides with this obstacle. Adding independent Gaussian noise to the actions from the standard RL policy can enable that policy to navigate around the obstacle, but only at the cost of entering the costly red states.

**Testing for different types of robustness.** Most prior work on robust RL focuses on changing static attributes of the environment, such as the mass or position of objects (Tessler et al., 2019; Kamalaruban et al., 2020). However, our analysis suggests that MaxEnt RL is robust against a wider range of perturbations, which we probe in our next set of experiments.

First, we introduced perturbations in the middle of an episode. We took the pushing task shown in Fig. 1 and, instead of adding an obstacle, perturbed the XY position of the puck after 20 time steps. By evaluating the reward of a policy while varying the size of this perturbation, we can study the range of disturbances to which MaxEnt RL is robust. We measured the performance of MaxEnt RL policies trained with different entropy coefficients $\alpha$. The results shown in Fig. 6 indicate all methods perform well on the environment without any disturbances, but only the MaxEnt RL trained with the largest entropy coefficient is robust to larger disturbances. This experiment supports our theory that the entropy coefficient determines the size of the robust set.

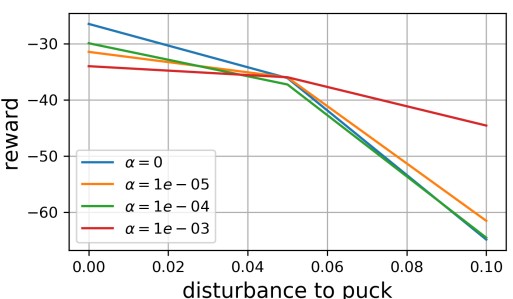

Figure 6: **Robustness to dynamic perturbations**: MaxEnt RL is robust to random external forces applied to the environment dynamics.

Our analysis suggests that MaxEnt RL is not only robust to random perturbations, but is actually robust against *adversarial* perturbations. We next compare MaxEnt RL and standard RL in this

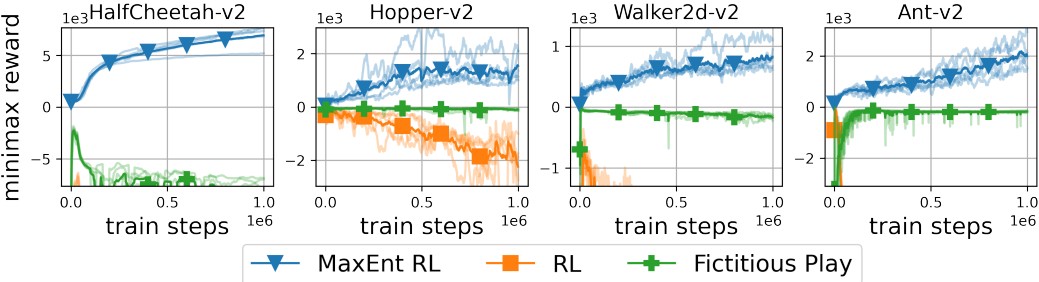

Figure 7: Robustness to adversarial perturbations of the environment dynamics.

Figure 8: MaxEnt RL policies are robust to disturbances in the reward function.

setting. We trained both algorithms on a peg insertion task shown in Fig. 7 *(left)*. Visualizing the learned policy, we observe that the standard RL policy always takes the same route to the goal, whereas the MaxEnt RL policy uses many different routes to get to the goal. For each policy we found the worst-case perturbation to the hole location using CMA-ES (Hansen, 2016). For small perturbations, both standard RL and MaxEnt RL achieve a success rate near 100%. However, for perturbations that are 1.5cm or 2cm in size, only MaxEnt RL continues to solve the task. For larger perturbations neither method can solve the task. In summary, this experiment highlights that MaxEnt RL is robust to *adversarial* disturbances, as predicted by the theory.

Finally, our analysis suggests that MaxEnt RL is also robust to perturbations to the reward function. To test this theoretical result, we apply MaxEnt RL on four continuous control tasks from the standard OpenAI Gym (Brockman et al., 2016) benchmark. We compare to SVG-0 (Heess et al., 2015) (which uses stochastic policy gradients) and to fictitious play (Brown, 1951). In the RL setting, fictitious play corresponds to modifying standard RL to use an adversarially-chosen reward function for each Bellman update. We evaluate the policy on an adversarially chosen reward function, chosen from the set defined in Equation 2. The analytic solution for this worst-case reward function is $\tilde{r}(\mathbf{s_t}, \mathbf{a_t}) = r(\mathbf{s_t}, \mathbf{a_t}) - \log \pi(\mathbf{a_t} \mid \mathbf{s_t})$. Both MaxEnt RL and standard RL can maximize the cumulative reward (see Fig. 11 in Appendix B), but only MaxEnt RL succeeds as maximizing the worst-case reward, as shown in Fig. 8. In summary, this experiment supports our proof that MaxEnt RL solves a robust RL problem for the set of rewards specified in Theorem 4.1.

## 6 DISCUSSION

In this paper, we formally showed that MaxEnt RL algorithms optimize a bound on a robust RL objective. This robust RL objective uses a different reward function than the MaxEnt RL objective. Our analysis characterizes the robust sets for both reward and dynamics perturbations, and provides intuition for how such algorithms should be used for robust RL problems. To our knowledge, our work is the first to formally characterize the robustness of MaxEnt RL algorithms, despite the fact that numerous papers have conjectured that such robustness results may be possible. Our experimental evaluation shows that, in line with our theoretical findings, simple MaxEnt RL algorithms perform competitively with (and sometimes better than) recently proposed adversarial robust RL methods on benchmarks proposed by those works.

Of course, MaxEnt RL methods are not necessarily the *ideal* approach to robustness: applying such methods still requires choosing a hyperparameter (the entropy coefficient), and the robust set for MaxEnt RL is not always simple. Nonetheless, we believe that the analysis in this paper, represents an important step towards a deeper theoretical understanding of the connections between robustness and entropy regularization in RL. We hope that this analysis will open the door for the development of new, simple algorithms for robust RL.

**Acknowledgments** We thank Ofir Nachum, Brendan O'Donoghue, Brian Ziebart, and anonymous reviewers for their feedback on an early drafts. BE is supported by the Fannie and John Hertz Foundation and the National Science Foundation GFRP (DGE 1745016).

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

## A    PROOFS

### A.1    USEFUL LEMMAS

Before stating the proofs, we recall and prove two (known) identities.

**Lemma A.1.**

$$E_{p(x,y)}[-\log p(x \mid y)] = \min_{f(x,y)} E_{p(x,y)}\left[-f(x,y) + \log \sum_{x'} e^{f(x',y)}\right]. \tag{7}$$

This identity says that the negative entropy function is the Fenchel dual of the log-sum-exp function. This identity can be proven using calculus of variations:

*Proof.* We start by finding the function $f(x,y)$ that optimizes the RHS of Eq. 7. Noting that the RHS is a convex function of $f(x,y)$, we can find the optimal $f(x,y)$ by taking the derivative and setting it equal to zero:

$$\frac{d}{df(x,y)}\left(E_{p(x,y)}[-f(x,y)] + \log \sum_{x'} e^{f(x',y)}\right) = -p(x,y) + \frac{e^{f(x,y)}}{\sum_{x'} e^{f(x',y)}}.$$

Setting this derivative equal to zero, we see that solution $f^*(x,y)$ satisfies

$$\frac{e^{f^*(x,y)}}{\sum_{x'} e^{f^*(x',y)}} = p(x,y).$$

Now, we observe that $p(y) = 1$:

$$p(y) = \sum_x p(x,y)$$

$$= \sum_x \frac{e^{f^*(x,y)}}{\sum_{x'} e^{f^*(x',y)}}$$

$$= \frac{\sum_x e^{f^*(x,y)}}{\sum_{x'} e^{f^*(x',y)}} = 1.$$

We then have

$$p(x \mid y) = \frac{p(x,y)}{p(y)}^{1} = \frac{e^{f^*(x,y)}}{\sum_{x'} e^{f^*(x',y)}}.$$

Taking the log of both sides, we have

$$\log p(x \mid y) = f^*(x,y) - \log \sum_{x'} e^{f^*(x',y)}.$$

Multiplying both sides by -1 and taking the expectation w.r.t. $p(x,y)$, we obtain the desired result:

$$E_{p(x,y)}[-\log p(x \mid y)] = E_{p(x,y)}[-f^*(x,y)] + \log \sum_{x'} e^{f^*(x',y)}$$

$$= \min_{f(x,y)} E_{p(x,y)}[-f(x,y)] + \log \sum_{x'} e^{f(x',y)}.$$

$\square$

This lemma also holds for functions with more variables, a result that will be useful when proving dynamics robustness.

**Lemma A.2.**

$$E_{p(x,y,z)}[-\log p(x,y \mid z)] = \min_{f(x,y,z)} E_{p(x,y,z)}\left[-f(x,y,z) + \log \sum_{x',y'} e^{f(x',y',z)}\right].$$

*Proof.* Simply apply Lemma A.1 to the random variables $(x,y)$ and $z$.    $\square$

## A.2 PROOF OF THEOREM 4.1

This section will provide a proof of Theorem 4.1.

*Proof.* We start by restating the objective for robust control with rewards:

$$\min_{\tilde{r} \in \tilde{\mathcal{R}}(\pi)} \mathbb{E}_{\substack{\mathbf{a_t} \sim \pi(\mathbf{a_t}|\mathbf{s_t}) \\ \mathbf{s_{t+1}} \sim p(\mathbf{s_{t+1}}|\mathbf{s_t},\mathbf{a_t})}} \left[ \sum_t \tilde{r}(\mathbf{s_t}, \mathbf{a_t}) \right]. \tag{8}$$

We now employ the KKT conditions (Boyd et al., 2004, Chpt. 5.5.3). If the robust set $\tilde{\mathcal{R}}$ is strictly feasibly, then there exists a constraint value $\epsilon \geq 0$ (the dual solution) such that the constrained problem is equivalent to the relaxed problem with Lagrange multiplier $\lambda = 1$:

$$\min_{\tilde{r}} \mathbb{E}_{\substack{\mathbf{a_t} \sim \pi(\mathbf{a_t}|\mathbf{s_t}) \\ \mathbf{s_{t+1}} \sim p(\mathbf{s_{t+1}}|\mathbf{s_t},\mathbf{a_t})}} \left[ \sum_t \tilde{r}(\mathbf{s_t}, \mathbf{a_t}) + \log \int_{\mathcal{A}} \exp(r(\mathbf{s_t}, \mathbf{a_t}) - \tilde{r}(\mathbf{s_t}, \mathbf{a_t})) d\mathbf{a_t}' \right].$$

To clarify exposition, we parameterize $\tilde{r}$ by its deviation from the original reward, $\Delta r(\mathbf{s_t}, \mathbf{a_t}) \triangleq r(\mathbf{s_t}, \mathbf{a_t}) - \tilde{r}(\mathbf{s_t}, \mathbf{a_t})$. We also define $\rho_t^{\pi}(\mathbf{s_t}, \mathbf{a_t})$ as the marginal distribution over states and actions visited by policy $\pi$ at time $t$. We now rewrite the relaxed objective in terms of $\Delta r(\mathbf{s_t}, \mathbf{a_t})$:

$$\min_{\Delta r} \mathbb{E}_{\substack{\mathbf{a_t} \sim \pi(\mathbf{a_t}|\mathbf{s_t}) \\ \mathbf{s_{t+1}} \sim p(\mathbf{s_{t+1}}|\mathbf{s_t},\mathbf{a_t})}} \left[ \sum_t r(\mathbf{s_t}, \mathbf{a_t}) - \Delta r(\mathbf{s_t}, \mathbf{a_t}) + \log \int_{\mathcal{A}} \exp(\Delta r(\mathbf{s_t}, \mathbf{a_t})) d\mathbf{a_t}' \right]$$

$$= \mathbb{E}_{\substack{\mathbf{a_t} \sim \pi(\mathbf{a_t}|\mathbf{s_t}) \\ \mathbf{s_{t+1}} \sim p(\mathbf{s_{t+1}}|\mathbf{s_t},\mathbf{a_t})}} \left[ \sum_t r(\mathbf{s_t}, \mathbf{a_t}) \right]$$

$$+ \min_{\Delta r} \mathbb{E}_{\substack{\mathbf{a_t} \sim \pi(\mathbf{a_t}|\mathbf{s_t}) \\ \mathbf{s_{t+1}} \sim p(\mathbf{s_{t+1}}|\mathbf{s_t},\mathbf{a_t})}} \left[ \sum_t -\Delta r(\mathbf{s_t}, \mathbf{a_t}) + \log \int_{\mathcal{A}} \exp(\Delta r(\mathbf{s_t}, \mathbf{a_t})) d\mathbf{a_t}' \right].$$

$$= \mathbb{E}_{\substack{\mathbf{a_t} \sim \pi(\mathbf{a_t}|\mathbf{s_t}) \\ \mathbf{s_{t+1}} \sim p(\mathbf{s_{t+1}}|\mathbf{s_t},\mathbf{a_t})}} \left[ \sum_t r(\mathbf{s_t}, \mathbf{a_t}) \right]$$

$$+ \min_{\Delta r} \sum_t \mathbb{E}_{\rho_t^{\pi}(\mathbf{s_t}, \mathbf{a_t})} \left[ -\Delta r(\mathbf{s_t}, \mathbf{a_t}) + \log \int_{\mathcal{A}} \exp(\Delta r(\mathbf{s_t}, \mathbf{a_t})) d\mathbf{a_t}' \right].$$

$$= \mathbb{E}_{\substack{\mathbf{a_t} \sim \pi(\mathbf{a_t}|\mathbf{s_t}) \\ \mathbf{s_{t+1}} \sim p(\mathbf{s_{t+1}}|\mathbf{s_t},\mathbf{a_t})}} \left[ \sum_t r(\mathbf{s_t}, \mathbf{a_t}) - \log \pi(\mathbf{a_t} \mid \mathbf{s_t}) \right].$$

The last line follows from applying Lemma A.1. $\qquad \square$

One important note about this proof is that the KKT conditions are applied to the problem of optimizing the reward function $\tilde{r}$, which is a convex optimization problem. We do not require that the policy $\pi(\mathbf{a_t} \mid \mathbf{s_t})$ or the dynamics $p(\mathbf{s_{t+1}} \mid \mathbf{s_t}, \mathbf{a_t})$ be convex.

## A.3 WORKED EXAMPLE OF REWARD ROBUSTNESS

We compute the penalty for Example 1 in Section 4.3 using a Gaussian integral:

$$\text{PENALTY}(\tilde{r}, \mathbf{s}) = \log \int_{-10}^{10} \exp(r(\mathbf{s}, \mathbf{a}) - \tilde{r}(\mathbf{s}, \mathbf{a})) d\mathbf{a}$$

$$= \log \int_{-10}^{10} \exp(-(\mathbf{a} - \mathbf{a}^*)^2 + \frac{1}{2}(\mathbf{a} - (\mathbf{a}^* + \Delta\mathbf{a}))^2) d\mathbf{a}$$

$$= \log \int_{-10}^{10} \exp(-\frac{1}{2}(\mathbf{a} - (\mathbf{a}^* + \Delta\mathbf{a}))^2 + \Delta\mathbf{a}^2) d\mathbf{a}$$

$$= \Delta\mathbf{a}^2 + \frac{1}{2} \log(2\pi) + \log(20).$$

## A.4 PROOF OF THEOREM 4.2

*Proof.* We now provide the proof of Theorem 4.2. The first step will be to convert the variation in dynamics into a sort of variation in the reward function. The second step will convert the constrained robust control objective into an unconstrained (penalized) objective. The third step will show that the penalty is equivalent to action entropy.

**Step 1: Dynamics variation → reward variation.** Our aim is to obtain a lower bound on the following objective:

$$\min_{\tilde{p}\in\tilde{\mathcal{P}}(\pi)} \mathbb{E}_{\substack{\mathbf{a_t}\sim\pi(\mathbf{a_t}|\mathbf{s_t}) \\ \mathbf{s_{t+1}}\sim\tilde{p}(\mathbf{s_{t+1}}|\mathbf{s_t},\mathbf{a_t})}} \left[\sum_t r(\mathbf{s_t},\mathbf{a_t})\right].$$

We start by taking a log-transform of this objective, noting that this transformation does not change the optimal policy as the log function is strictly monotone increasing.

$$\log\mathbb{E}_{\substack{\mathbf{a_t}\sim\pi(\mathbf{a_t}|\mathbf{s_t}) \\ \mathbf{s_{t+1}}\sim\tilde{p}(\mathbf{s_{t+1}}|\mathbf{s_t},\mathbf{a_t})}} \left[\sum_t r(\mathbf{s_t},\mathbf{a_t})\right].$$

Note that we assumed that the reward was positive, so the logarithm is well defined. We can write this objective in terms of the adversarial dynamics using importance weights.

$$\log\mathbb{E}_{\substack{\mathbf{a_t}\sim\pi(\mathbf{a_t}|\mathbf{s_t}) \\ \mathbf{s_{t+1}}\sim p(\mathbf{s_{t+1}}|\mathbf{s_t},\mathbf{a_t})}} \left[\left(\prod_t\frac{\tilde{p}(\mathbf{s_{t+1}}\mid\mathbf{s_t},\mathbf{a_t})}{p(\mathbf{s_{t+1}}\mid\mathbf{s_t},\mathbf{a_t})}\right)\sum_t r(\mathbf{s_t},\mathbf{a_t})\right]$$

$$=\log\mathbb{E}_{\substack{\mathbf{a_t}\sim\pi(\mathbf{a_t}|\mathbf{s_t}) \\ \mathbf{s_{t+1}}\sim p(\mathbf{s_{t+1}}|\mathbf{s_t},\mathbf{a_t})}} \left[\exp\left(\log\left(\sum_t r(\mathbf{s_t},\mathbf{a_t})\right) + \sum_t\log\tilde{p}(\mathbf{s_{t+1}}\mid\mathbf{s_t},\mathbf{a_t}) - \log p(\mathbf{s_{t+1}}\mid\mathbf{s_t},\mathbf{a_t})\right)\right]$$

$$\overset{(a)}{\geq}\mathbb{E}_{\substack{\mathbf{a_t}\sim\pi(\mathbf{a_t}|\mathbf{s_t}) \\ \mathbf{s_{t+1}}\sim p(\mathbf{s_{t+1}}|\mathbf{s_t},\mathbf{a_t})}} \left[\log\left(\sum_t r(\mathbf{s_t},\mathbf{a_t})\right) + \sum_t\log\tilde{p}(\mathbf{s_{t+1}}\mid\mathbf{s_t},\mathbf{a_t}) - \log p(\mathbf{s_{t+1}}\mid\mathbf{s_t},\mathbf{a_t})\right]$$

$$=\mathbb{E}_{\substack{\mathbf{a_t}\sim\pi(\mathbf{a_t}|\mathbf{s_t}) \\ \mathbf{s_{t+1}}\sim p(\mathbf{s_{t+1}}|\mathbf{s_t},\mathbf{a_t})}} \left[\log\left(\frac{1}{T}\sum_t r(\mathbf{s_t},\mathbf{a_t})\right) + \log T + \sum_t\log\tilde{p}(\mathbf{s_{t+1}}\mid\mathbf{s_t},\mathbf{a_t}) - \log p(\mathbf{s_{t+1}}\mid\mathbf{s_t},\mathbf{a_t})\right]$$

$$\overset{(b)}{\geq}\mathbb{E}_{\substack{\mathbf{a_t}\sim\pi(\mathbf{a_t}|\mathbf{s_t}) \\ \mathbf{s_{t+1}}\sim p(\mathbf{s_{t+1}}|\mathbf{s_t},\mathbf{a_t})}} \left[\sum_t\frac{1}{T}\log r(\mathbf{s_t},\mathbf{a_t}) + \log\tilde{p}(\mathbf{s_{t+1}}\mid\mathbf{s_t},\mathbf{a_t}) - \log p(\mathbf{s_{t+1}}\mid\mathbf{s_t},\mathbf{a_t})\right] + \log T. \quad (9)$$

Both inequalities are applications of Jensen's inequality. As before, our assumption that the rewards are positive ensures that the logarithms remain well defined. While the adversary is choosing the dynamics under which we will evaluate the policy, we are optimizing a lower bound which depends on a *different* dynamics function. *This step allows us to analyze adversarially-chosen dynamics as perturbations to the reward.*

**Step 2: Relaxing the constrained objective** To clarify exposition, we will parameterize the adversarial dynamics as a deviation from the true dynamics:

$$\Delta r(\mathbf{s_{t+1}},\mathbf{s_t},\mathbf{a_t}) = \log p(\mathbf{s_{t+1}}\mid\mathbf{s_t},\mathbf{a_t}) - \log\tilde{p}(\mathbf{s_{t+1}}\mid\mathbf{s_t},\mathbf{a_t}). \quad (10)$$

The constraint that the adversarial dynamics integrate to one can be expressed as

$$\int_{\mathcal{S}}\underbrace{p(\mathbf{s_{t+1}}\mid\mathbf{s_t},\mathbf{a_t})e^{-\Delta r(\mathbf{s_{t+1}},\mathbf{s_t},\mathbf{a_t})}}_{\tilde{p}(\mathbf{s_{t+1}}|\mathbf{s_t},\mathbf{a_t})}\,d\mathbf{s_{t+1}} = 1$$

Using this notation, we can write the lower bound on the robust control problem (Eq. 9) as follows:

$$\min_{\Delta r}\mathbb{E}_{\substack{\mathbf{a_t}\sim\pi(\mathbf{a_t}|\mathbf{s_t}), \\ \mathbf{s_{t+1}}\sim p(\mathbf{s_{t+1}}|\mathbf{s_t},\mathbf{a_t})}} \left[\sum_t\frac{1}{T}\log r(\mathbf{s_t},\mathbf{a_t}) - \Delta r(\mathbf{s_{t+1}},\mathbf{s_t},\mathbf{a_t})\right] + \log T \quad (11)$$

$$\text{s.t.}\quad \mathbb{E}_{\substack{\mathbf{a_t}\sim\pi(\mathbf{a_t}|\mathbf{s_t}), \\ \mathbf{s_{t+1}}\sim p(\mathbf{s_{t+1}}|\mathbf{s_t},\mathbf{a_t})}} \left[\sum_t\log\iint_{\mathcal{A}\times\mathcal{S}}e^{\Delta r(\mathbf{s_{t+1}}',\mathbf{s_t},\mathbf{a_t}')}d\mathbf{a_t}'d\mathbf{s_{t+1}}'\right] \leq \epsilon \quad (12)$$

$$\text{and}\quad \int_{\mathcal{S}}\underbrace{p(\mathbf{s_{t+1}}\mid\mathbf{s_t},\mathbf{a_t})e^{-\Delta r(\mathbf{s_{t+1}},\mathbf{s_t},\mathbf{a_t})}}_{\tilde{p}(\mathbf{s_{t+1}}|\mathbf{s_t},\mathbf{a_t})}\,d\mathbf{s_{t+1}} = 1 \quad \forall\mathbf{s_t},\mathbf{a_t}. \quad (13)$$

The constraint in Eq. 12 is the definition of the set of adversarial dynamics $\tilde{\mathcal{P}}$, and the constraint in Eq. 13 ensures that $\tilde{p}(\mathbf{s_{t+1}}\mid\mathbf{s_t},\mathbf{a_t})$ represents a valid probability density function.

Note that $\Delta r(\mathbf{s_{t+1}},\mathbf{s_t},\mathbf{a_t}) = 0$ is a *strictly feasible* solution to this constrained optimization problem for $\epsilon > 0$. Note also that the problem of optimizing the function $\Delta r(\mathbf{s_t},\mathbf{a_t})$ is a convex optimization

problem. We can therefore employ the KKT conditions (Boyd et al., 2004, Chpt. 5.5.3). If the robust set $\tilde{\mathcal{P}}$ is strictly feasibly, then there exists $\epsilon \geq 0$ (the dual solution) such that the set of solutions $\tilde{p}$ to the constrained optimization problem Eq. 11 are equivalent to the set of solutions to the following relaxed objective with Lagrange multiplier $\lambda = 1$:

$$\min_{\Delta r} \mathbb{E}_{\substack{\mathbf{a_t} \sim \pi(\mathbf{a_t}|\mathbf{s_t}), \\ \mathbf{s_{t+1}} \sim p(\mathbf{s_{t+1}}|\mathbf{s_t}, \mathbf{a_t})}} \left[ \sum_t \frac{1}{T} \log r(\mathbf{s_t}, \mathbf{a_t}) - \Delta r(\mathbf{s_{t+1}}, \mathbf{s_t}, \mathbf{a_t}) \right. \tag{14}$$
$$\left. + \log \iint_{\mathcal{A} \times \mathcal{S}} e^{\Delta r(\mathbf{s_{t+1}}', \mathbf{s_t}, \mathbf{a_t}')} d\mathbf{a_t}' d\mathbf{s_{t+1}}' \right] + \log T$$

$$\text{s.t.} \quad \int_{\mathcal{S}} p(\mathbf{s_{t+1}} \mid \mathbf{s_t}, \mathbf{a_t}) e^{-\Delta r(\mathbf{s_{t+1}}', \mathbf{s_t}, \mathbf{a_t}')} d\mathbf{s_{t+1}} = 1 \quad \forall \mathbf{s_t}, \mathbf{a_t}. \tag{15}$$

This step does not require that the policy $\pi(\mathbf{a_t} \mid \mathbf{s_t})$ or the dynamics $p(\mathbf{s_{t+1}} \mid \mathbf{s_t}, \mathbf{a_t})$ be convex.

Our next step is to show that the constraint does not affect the solution of this optimization problem. For any function $\Delta r(\mathbf{s_{t+1}}, \mathbf{s_t}, \mathbf{a_t})$, we can add a constant $c(\mathbf{s_t}, \mathbf{a_t})$ and obtain the same objective value but now satisfy the constraint. We construct $c(\mathbf{s_t}, \mathbf{a_t})$ as

$$c(\mathbf{s_t}, \mathbf{a_t}) = \log \int_{\mathcal{S}} p(\mathbf{s_{t+1}} \mid \mathbf{s_t}, \mathbf{a_t}) e^{-\Delta r(\mathbf{s_{t+1}}', \mathbf{s_t}, \mathbf{a_t}')} d\mathbf{s_{t+1}} = 1 \quad \forall \mathbf{s_t}, \mathbf{a_t}.$$

First, we observe that adding $c(\mathbf{s_t}, \mathbf{a_t})$ to $\Delta r$ does not change the objective:

$$\mathbb{E}_{\substack{\mathbf{a_t} \sim \pi(\mathbf{a_t}|\mathbf{s_t}), \\ \mathbf{s_{t+1}} \sim p(\mathbf{s_{t+1}}|\mathbf{s_t}, \mathbf{a_t})}} \left[ \sum_t \frac{1}{T} \log r(\mathbf{s_t}, \mathbf{a_t}) - (\Delta r(\mathbf{s_{t+1}}, \mathbf{s_t}, \mathbf{a_t}) + c(\mathbf{s_t}, \mathbf{a_t})) \right.$$
$$\left. + \log \iint_{\mathcal{A} \times \mathcal{S}} e^{\Delta r(\mathbf{s_{t+1}}', \mathbf{s_t}, \mathbf{a_t}') + c(\mathbf{s_t}, \mathbf{a_t})} d\mathbf{a_t}' d\mathbf{s_{t+1}}' \right] + \log T$$

$$= \mathbb{E}_{\substack{\mathbf{a_t} \sim \pi(\mathbf{a_t}|\mathbf{s_t}), \\ \mathbf{s_{t+1}} \sim p(\mathbf{s_{t+1}}|\mathbf{s_t}, \mathbf{a_t})}} \left[ \sum_t \frac{1}{T} \log r(\mathbf{s_t}, \mathbf{a_t}) - \Delta r(\mathbf{s_{t+1}}, \mathbf{s_t}, \mathbf{a_t}) - c(\mathbf{s_t}, \mathbf{a_t}) \right.$$
$$\left. + \log \left( e^{c(\mathbf{s_t}, \mathbf{a_t})} \iint_{\mathcal{A} \times \mathcal{S}} e^{\Delta r(\mathbf{s_{t+1}}', \mathbf{s_t}, \mathbf{a_t}')} d\mathbf{a_t}' d\mathbf{s_{t+1}}' \right) \right] + \log T$$

$$= \mathbb{E}_{\substack{\mathbf{a_t} \sim \pi(\mathbf{a_t}|\mathbf{s_t}), \\ \mathbf{s_{t+1}} \sim p(\mathbf{s_{t+1}}|\mathbf{s_t}, \mathbf{a_t})}} \left[ \sum_t \frac{1}{T} \log r(\mathbf{s_t}, \mathbf{a_t}) - \Delta r(\mathbf{s_{t+1}}, \mathbf{s_t}, \mathbf{a_t}) - \cancel{c(\mathbf{s_t}, \mathbf{a_t})} + \cancel{c(\mathbf{s_t}, \mathbf{a_t})} \right.$$
$$\left. + \log \left( \iint_{\mathcal{A} \times \mathcal{S}} e^{\Delta r(\mathbf{s_{t+1}}', \mathbf{s_t}, \mathbf{a_t}')} d\mathbf{a_t}' d\mathbf{s_{t+1}}' \right) \right] + \log T.$$

Second, we observe that the new reward function $\Delta r(\mathbf{s_{t+1}}, \mathbf{s_t}, \mathbf{a_t}) + c(\mathbf{s_t}, \mathbf{a_t})$ satisfies the constraint in Eq. 15:

$$\int_{\mathcal{S}} p(\mathbf{s_{t+1}} \mid \mathbf{s_t}, \mathbf{a_t}) e^{-(\Delta r(\mathbf{s_{t+1}}', \mathbf{s_t}, \mathbf{a_t}') - c(\mathbf{s_t}, \mathbf{a_t}))} d\mathbf{s_{t+1}}$$
$$= e^{-c(\mathbf{s_t}, \mathbf{a_t})} \int_{\mathcal{S}} p(\mathbf{s_{t+1}} \mid \mathbf{s_t}, \mathbf{a_t}) e^{-\Delta r(\mathbf{s_{t+1}}', \mathbf{s_t}, \mathbf{a_t}')} d\mathbf{s_{t+1}}$$
$$= \frac{\int_{\mathcal{S}} p(\mathbf{s_{t+1}} \mid \mathbf{s_t}, \mathbf{a_t}) e^{-\Delta r(\mathbf{s_{t+1}}', \mathbf{s_t}, \mathbf{a_t}')} d\mathbf{s_{t+1}}}{\int_{\mathcal{S}} p(\mathbf{s_{t+1}} \mid \mathbf{s_t}, \mathbf{a_t}) e^{-\Delta r(\mathbf{s_{t+1}}', \mathbf{s_t}, \mathbf{a_t}')} d\mathbf{s_{t+1}}} = 1.$$

Thus, constraining $\Delta r$ to represent a probability distribution does not affect the solution (value) to the optimization problem, so we can ignore this constraint without loss of generality. The new, unconstrained objective is

$$\min_{\Delta r} \mathbb{E}_{\substack{a \sim \pi(\mathbf{a_t}|\mathbf{s_t}), \\ s' \sim p(\mathbf{s_{t+1}}|\mathbf{s_t}, \mathbf{a_t})}} \left[ \sum_t \frac{1}{T} \log r(\mathbf{s_t}, \mathbf{a_t}) - \Delta r(\mathbf{s_{t+1}}, \mathbf{s_t}, \mathbf{a_t}) \right.$$
$$\left. + \log \iint_{\mathcal{A} \times \mathcal{S}} e^{\Delta r(\mathbf{s_{t+1}}, \mathbf{s_t}, \mathbf{a_t})} d\mathbf{a_t}' d\mathbf{s_{t+1}}' \right] + \log T. \tag{16}$$

**Step 3: The penalty is the Fenchel dual of action entropy.** We define $\rho_t^\pi(\mathbf{s_t}, \mathbf{a_t}, \mathbf{s_{t+1}})$ as the marginal distribution of transitions visited by policy $\pi$ at time $t$. We now apply Lemma A.2 to Eq. 16:

$$\min_{\Delta r} \mathbb{E}_{\substack{a \sim \pi(\mathbf{a_t}|\mathbf{s_t}), \\ s' \sim p(\mathbf{s_{t+1}}|\mathbf{s_t},\mathbf{a_t})}} \left[ \sum_t \frac{1}{T} \log r(\mathbf{s_t}, \mathbf{a_t}) - \Delta r(\mathbf{s_{t+1}}, \mathbf{s_t}, \mathbf{a_t}) \right.$$
$$\left. + \log \iint_{\mathcal{A} \times \mathcal{S}} e^{\Delta r(\mathbf{s_{t+1}}, \mathbf{s_t}', \mathbf{a_t}')} d\mathbf{a_t}' d\mathbf{s_{t+1}}' \right] + \log T \quad (17)$$

$$= \mathbb{E}_{\substack{a \sim \pi(\mathbf{a_t}|\mathbf{s_t}), \\ s' \sim p(\mathbf{s_{t+1}}|\mathbf{s_t},\mathbf{a_t})}} \left[ \sum_t \frac{1}{T} \log r(\mathbf{s_t}, \mathbf{a_t}) \right] + \log T$$
$$+ \min_{\Delta r} \mathbb{E}_{\substack{a \sim \pi(\mathbf{a_t}|\mathbf{s_t}), \\ s' \sim p(\mathbf{s_{t+1}}|\mathbf{s_t},\mathbf{a_t})}} \left[ \sum_t -\Delta r(\mathbf{s_{t+1}}, \mathbf{s_t}, \mathbf{a_t}) + \log \iint_{\mathcal{A} \times \mathcal{S}} e^{\Delta r(\mathbf{s_{t+1}}, \mathbf{s_t}', \mathbf{a_t}')} d\mathbf{a_t}' d\mathbf{s_{t+1}}' \right]$$

$$= \mathbb{E}_{\substack{a \sim \pi(\mathbf{a_t}|\mathbf{s_t}), \\ s' \sim p(\mathbf{s_{t+1}}|\mathbf{s_t},\mathbf{a_t})}} \left[ \sum_t \frac{1}{T} \log r(\mathbf{s_t}, \mathbf{a_t}) \right] + \log T$$
$$+ \min_{\Delta r} \sum_t \mathbb{E}_{\rho_t^\pi(\mathbf{s_t}, \mathbf{a_t}, \mathbf{s_{t+1}})} \left[ -\Delta r(\mathbf{s_{t+1}}, \mathbf{s_t}, \mathbf{a_t}) + \log \iint_{\mathcal{A} \times \mathcal{S}} e^{\Delta r(\mathbf{s_{t+1}}, \mathbf{s_t}', \mathbf{a_t}')} d\mathbf{a_t}' d\mathbf{s_{t+1}}' \right]$$

$$= \mathbb{E}_{\substack{a \sim \pi(\mathbf{a_t}|\mathbf{s_t}), \\ s' \sim p(\mathbf{s_{t+1}}|\mathbf{s_t},\mathbf{a_t})}} \left[ \sum_t \frac{1}{T} \log r(\mathbf{s_t}, \mathbf{a_t}) - \log p(\mathbf{a_t}, \mathbf{s_{t+1}} \mid \mathbf{s_t}) \right] + \log T$$

$$= \mathbb{E}_{\substack{a \sim \pi(\mathbf{a_t}|\mathbf{s_t}), \\ s' \sim p(\mathbf{s_{t+1}}|\mathbf{s_t},\mathbf{a_t})}} \left[ \sum_t \frac{1}{T} \log r(\mathbf{s_t}, \mathbf{a_t}) - \log \pi(\mathbf{a_t} \mid \mathbf{s_t}) - \log p(\mathbf{s_{t+1}} \mid \mathbf{s_t}, \mathbf{a_t}) \right] + \log T$$

**Summary.** We have thus shown the follow:
$$\min_{\tilde{p} \in \tilde{\mathcal{P}}} \log J_{\text{MaxEnt}}(\pi; \tilde{p}, r) \geq J_{\text{MaxEnt}}(\pi; p, \bar{r}) + \log T.$$

Taking the exponential transform of both sides, we obtain the desired result.

$\square$

## A.5 Robustness to Both Rewards and Dynamics

While Theorem 4.2 is phrased in terms of perturbations to the dynamics function, not the reward function, we now discuss how this result can be used to show that MaxEnt RL is simultaneously robust to perturbations in the dynamics and the reward function. Define a modified MDP where the reward is appended to the observation. Then the reward function is the last coordinate of the observation. In this scenario, robustness to dynamics is equivalent to robustness to rewards.

## A.6 How Big is the Robust Set ($\epsilon$)?

Our proof of Theorem 4.2 used duality to argue that there exists an $\epsilon$ for which MaxEnt RL maximizes a lower bound on the robust RL objective. However, we did not specify the size of this $\epsilon$. This raises the concern that $\epsilon$ might be arbitrarily small, even zero, in which case the result would be vacuous. In this section we provide a proof of Lemma 4.3, which provides a lower bound on the size of the robust set.

*Proof.* Our proof proceeds in three steps. We first argue that, at optimality, the constraint on the adversarial dynamics is tight. This will allow us to treat the constraint as an equality constraint, rather than an inequality constraint. Second, since this constraint holds with equality, then we can rearrange the constraint and solve for $\epsilon$ in terms of the optimal adversarial dynamics. The third step is to simplify the expression for $\epsilon$.

**Step 1: The constraint on $\Delta r$ holds with equality.** Our aim here is to show that the constraint on $\Delta r$ in Eq. 12 holds with equality for the optimal $\Delta r$ (i.e., that which optimizes Eq. 11).[1] The objective

---

[1] In the case where there are multiple optimal $\Delta r$, we require that the constraint hold with equality for at least one of the optimal $\Delta r$.

in Eq. 11 is linear in $\Delta r$, so there must exist optimal $\Delta r$ at the boundary of the constraint (Rockafellar, 1970, Chapter 32).

**Step 2: Solving for $\epsilon$.** Since the solution to the constrained optimization problem in Eq. 11 occurs at the boundary, the constraint in Eq. 12 holds with equality, immediately telling us the value of $\epsilon$:

$$
\begin{aligned}
\epsilon &= \mathbb{E}_{\substack{\mathbf{a_t} \sim \pi(\mathbf{a_t}|\mathbf{s_t}), \\ \mathbf{s_{t+1}} \sim p(\mathbf{s_{t+1}}|\mathbf{s_t}, \mathbf{a_t})}} \left[ \sum_t \log \iint_{\mathcal{A} \times \mathcal{S}} e^{\Delta r(\mathbf{s_{t+1}}', \mathbf{s_t}, \mathbf{a_t}')} d\mathbf{a_t}' d\mathbf{s_{t+1}}' \right] \\
&= T \cdot \mathbb{E}_{\rho(\mathbf{s_t})} \left[ \log \iint_{\mathcal{A} \times \mathcal{S}} e^{\Delta r(\mathbf{s_{t+1}}', \mathbf{s_t}, \mathbf{a_t}')} d\mathbf{a_t}' d\mathbf{s_{t+1}}' \right],
\end{aligned}
\tag{18}
$$

where $\Delta r$ is the solution to Eq. 17. This identity holds for all states $\mathbf{s_t}$. The second line above expresses the expectation over trajectories as an expectation over states, which will simplify the analysis in the rest of this proof. The factor of $T$ is introduced because we have removed the inner summation.

**Step 3: Simplifying the expression for $\epsilon$.** To better understand this value of $\epsilon$, we recall that the following identity (Lemma A.2) holds for this optimal $\Delta r$:

$$
\frac{e^{\Delta r(\mathbf{s_t}, \mathbf{a_t}, \mathbf{s_{t+1}})}}{\iint_{\mathcal{A} \times \mathcal{S}} e^{\Delta r(\mathbf{s_t}, \mathbf{a_t}', \mathbf{s_{t+1}}')} d\mathbf{a_t}', d\mathbf{s_{t+1}}'} = \rho(\mathbf{a_t}, \mathbf{s_{t+1}} \mid \mathbf{s_t}) \qquad \forall \mathbf{s_t}, \mathbf{a_t}, \mathbf{s_{t+1}}.
$$

We next take the $\log(\cdot)$ of both sides and rearrange terms:

$$
\log \iint_{\mathcal{A} \times \mathcal{S}} e^{\Delta r(\mathbf{s_t}, \mathbf{a_t}', \mathbf{s_{t+1}}')} d\mathbf{a_t}', d\mathbf{s_{t+1}}' = \Delta r(\mathbf{s_t}, \mathbf{a_t}, \mathbf{s_{t+1}}) - \log \rho(\mathbf{a_t}, \mathbf{s_{t+1}} \mid \mathbf{s_t}).
$$

Next, we substitute the definition of $\Delta r$ (Eq. 10) and factor $\log \rho(\mathbf{a_t}, \mathbf{s_{t+1}} \mid \mathbf{s_t})$:

$$
\begin{aligned}
\log &\iint_{\mathcal{A} \times \mathcal{S}} e^{\Delta r(\mathbf{s_t}, \mathbf{a_t}', \mathbf{s_{t+1}}')} d\mathbf{a_t}', d\mathbf{s_{t+1}}' \\
&= \log p(\mathbf{s_{t+1}} \mid \mathbf{s_t}, \mathbf{a_t}) - \log \tilde{p}(\mathbf{s_{t+1}} \mid \mathbf{s_t}, \mathbf{a_t}) - \log p(\mathbf{s_{t+1}} \mid \mathbf{s_t}, \mathbf{a_t}) - \log \pi(\mathbf{a_t} \mid \mathbf{s_t}) \\
&= -\log \tilde{p}(\mathbf{s_{t+1}} \mid \mathbf{s_t}, \mathbf{a_t}) - \log \pi(\mathbf{a_t} \mid \mathbf{s_t}).
\end{aligned}
$$

Substituting this expression into Eq. 18, we obtain a more intuitive expression for $\epsilon$:

$$
\begin{aligned}
\epsilon &= T \cdot \mathbb{E}_{\substack{\mathbf{a_t} \sim \pi(\mathbf{a_t}|\mathbf{s_t}), \\ \mathbf{s_{t+1}} \sim p(\mathbf{s_{t+1}}|\mathbf{s_t}, \mathbf{a_t})}} \left[ -\log \tilde{p}(\mathbf{s_{t+1}} \mid \mathbf{s_t}, \mathbf{a_t}) - \log \pi(\mathbf{a_t} \mid \mathbf{s_t}) \right] \\
&= T \cdot \mathbb{E}_{\substack{\mathbf{a_t} \sim \pi(\mathbf{a_t}|\mathbf{s_t}), \\ \mathbf{s_{t+1}} \sim p(\mathbf{s_{t+1}}|\mathbf{s_t}, \mathbf{a_t})}} \left[ \mathcal{H}_{\tilde{p}}[\mathbf{s_{t+1}} \mid \mathbf{s_t}, \mathbf{a_t}] + \mathcal{H}_\pi[\mathbf{a_t} \mid \mathbf{s_t}] \right] \\
&\geq T \cdot \mathbb{E}_{\substack{\mathbf{a_t} \sim \pi(\mathbf{a_t}|\mathbf{s_t}), \\ \mathbf{s_{t+1}} \sim p(\mathbf{s_{t+1}}|\mathbf{s_t}, \mathbf{a_t})}} \left[ \mathcal{H}_\pi[\mathbf{a_t} \mid \mathbf{s_t}] \right].
\end{aligned}
$$

The last line follows from our assumption that the state space is discrete, so the entropy $\mathcal{H}_{\tilde{p}}[\mathbf{s_{t+1}} \mid \mathbf{s_t}, \mathbf{a_t}]$ is non-negative. This same result will hold in environments with continuous state spaces as long as the (differential) entropy $\mathcal{H}_{\tilde{p}}[\mathbf{s_{t+1}} \mid \mathbf{s_t}, \mathbf{a_t}]$ is non-negative. $\qquad \square$

### A.7 WORKED EXAMPLE OF DYNAMICS ROBUSTNESS

We calculate the penalty using a Gaussian integral:

$$
\begin{aligned}
\log &\iint_{\mathcal{A} \times \mathcal{S}} \frac{p(\mathbf{s_{t+1}} \mid \mathbf{s_t}, \mathbf{a_t})}{\tilde{p}(\mathbf{s_{t+1}} \mid \mathbf{s_t}, \mathbf{a_t})} d\mathbf{a_t} d\mathbf{s_{t+1}} \\
&= \log \iint_{\mathcal{A} \times \mathcal{S}} 2 \exp\left(-\frac{1}{2}(\mathbf{s_{t+1}} - (A\mathbf{s_t} + B\mathbf{a_t}))^2 + \frac{1}{4}(\mathbf{s_{t+1}} - (A\mathbf{s_t} + B\mathbf{a_t} - \beta))^2\right) d\mathbf{a_t} d\mathbf{s_{t+1}} \\
&= \log \iint_{\mathcal{A} \times \mathcal{S}} 2 \exp\left(-\frac{1}{4}(\mathbf{s_{t+1}} - (A\mathbf{s_t} + B\mathbf{a_t} - \beta))^2 + \frac{1}{2}\beta^2\right) d\mathbf{a_t} d\mathbf{s_{t+1}} \\
&= \log \left(2\sqrt{4\pi} \int_{-10}^{10} \exp(\frac{1}{2}\beta^2) d\mathbf{a_t}\right) \\
&= \frac{1}{2}\beta^2 + \log(8\sqrt{\pi}) + \log(20).
\end{aligned}
$$

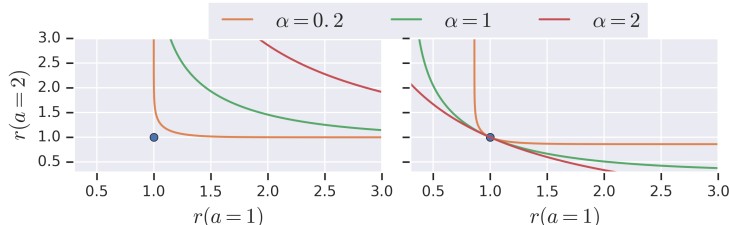

Figure 9: **Effect of Temperature**: *(Left)* For a given reward function (blue dot), we plot the robust sets for various values of the temperature. Somewhat surprisingly, it appears that increasing the temperature decreases the set of reward functions that MaxEnt is robust against. *(Right)* We examine the opposite: for a given reward function, which other robust sets might contain this reward function. We observe that robust sets corresponding to larger temperatures (i.e., the red curve) can be simultaneously robust against more reward functions than robust sets at lower temperatures.

We make two observations about this example. First, the penalty would be infinite if the adversary dynamics $\tilde{p}$ had the same variance as the true dynamics, $p$: the adversary must choose dynamics with higher variance. Second, the penalty depends on the size of the state space. More precisely, the penalty depends on the region over which the adversary applies their perturbation. The adversary can incur a smaller penalty by applying the perturbation over a smaller range of states.

## A.8 TEMPERATURES

Many algorithms for MaxEnt RL (Haarnoja et al., 2018a; Fox et al., 2016; Nachum et al., 2017) include a temperature $\alpha > 0$ to balance the reward and entropy terms:

$$J_{\text{MaxEnt}}(\pi, r) = \mathbb{E}_\pi \left[ \sum_{t=1}^T r(\mathbf{s_t}, \mathbf{a_t}) \right] + \alpha \mathcal{H}_\pi [\mathbf{a_t} \mid \mathbf{s_t}].$$

We can gain some intuition into the effect of this temperature on the set of reward functions to which we are robust. In particular, including a temperature $\alpha$ results in the following robust set:

$$R_r^\alpha = \left\{ r(\mathbf{s_t}, \mathbf{a_t}) + \alpha u_{\mathbf{s_t}}(\mathbf{a_t}) \mid u_{\mathbf{s_t}}(\mathbf{a_t}) \geq 0 \; \forall \mathbf{s_t}, \mathbf{a_t} \text{ and } \int_\mathcal{A} e^{-u_{\mathbf{s_t}}(\mathbf{a_t})} d\mathbf{a_t} \leq 1 \; \forall \mathbf{s_t} \right\}$$

$$= \left\{ r(\mathbf{s_t}, \mathbf{a_t}) + u_{\mathbf{s_t}}(\mathbf{a_t}) \mid u_{\mathbf{s_t}}(\mathbf{a_t}) \geq 0 \; \forall \mathbf{s_t}, \mathbf{a_t} \text{ and } \int_\mathcal{A} e^{-u_{\mathbf{s_t}}(\mathbf{a_t})/\alpha} d\mathbf{a_t} \leq 1 \; \forall \mathbf{s_t} \right\}. \quad (19)$$

In the second line, we simply moved the temperature from the objective to the constraint by redefining $u_{\mathbf{s_t}}(\mathbf{a_t}) \to \frac{1}{\alpha} u_{\mathbf{s_t}}(\mathbf{a_t})$.

We visualize the effect of the temperature in Fig. 9. First, we fix a reward function $r$, and plot the robust set $R_r^\alpha$ for varying values of $\alpha$. Fig. 9 (left) shows the somewhat surprising result that increasing the temperature (i.e., putting more weight on the entropy term) makes the policy *less* robust. In fact, the robust set for higher temperatures is a strict subset of the robust set for lower temperatures:

$$\alpha_1 < \alpha_2 \implies R_r^{\alpha_2} \subseteq R_r^{\alpha_2}.$$

This statement can be proven by simply noting that the function $e^{-\frac{x}{\alpha}}$ is an increasing function of $\alpha$ in Equation 19. It is important to recognize that being robust against more reward functions is not always desirable. In many cases, to be robust to everything, an optimal policy must do nothing.

We now analyze the temperature in terms of the converse question: if a reward function $r'$ is included in a robust set, what other reward functions are included in that robust set? To do this, we take a reward function $r'$, and find robust sets $R_r^\alpha$ that include $r'$, for varying values of $\alpha$. As shown in Fig. 9 (right), if we must be robust to $r'$ and use a high temperature, the only other reward functions to which we are robust are those that are similar, or pointwise weakly better, than $r'$. In contrast, when using a small temperature, we are robust against a wide range of reward functions, including those that are highly dissimilar from our original reward function (i.e., have higher reward for some actions, lower reward for other actions). Intuitively, increasing the temperature allows us to simultaneously be robust to a larger set of reward functions.

## A.9 MAXENT SOLVES ROBUST CONTROL FOR REWARDS

In Sec. 4.1, we showed that MaxEnt RL is equivalent to some robust-reward problem. The aim of this section is to go backwards: given a set of reward functions, can we formulate a MaxEnt RL problem such that the robust-reward problem and the MaxEnt RL problem have the same solution?

**Lemma A.3.** *For any collection of reward functions $R$, there exists another reward function $r$ such that the MaxEnt RL policy w.r.t. $r$ is an optimal robust-reward policy for $R$:*

$$\arg\max_{\pi} \mathbb{E}_{\pi}\left[\sum_{t=1}^{T} r(\mathbf{s_t}, \mathbf{a_t})\right] + \mathcal{H}_{\pi}[a \mid s] \subseteq \arg\max_{\pi} \min_{r' \in R} \mathbb{E}_{\pi}\left[\sum_{t=1}^{T} r'(\mathbf{s_t}, \mathbf{a_t})\right].$$

We use set containment, rather than equality, because there may be multiple solutions to the robust-reward control problem.

*Proof.* Let $\pi^*$ be a solution to the robust-reward control problem:

$$\pi^* \in \arg\max_{\pi} \min_{r_i \in R} \mathbb{E}_{\pi}\left[\sum_{t=1}^{T} r_i(\mathbf{s_t}, \mathbf{a_t})\right].$$

Define the MaxEnt RL reward function as follows:

$$r(\mathbf{s_t}, \mathbf{a_t}) = \log \pi^*(\mathbf{a_t} \mid \mathbf{s_t}).$$

Substituting this reward function in Equation 1, we see that the unique solution is $\pi = \pi^*$. $\qquad\square$

Intuitively, this theorem states that we can use MaxEnt RL to solve *any* robust-reward control problem that requires robustness with respect to any arbitrary set of rewards, if we can find the right corresponding reward function $r$ for MaxEnt RL. One way of viewing this theorem is as providing an avenue to sidestep the challenges of robust-reward optimization. Unfortunately, we will still have to perform robust optimization to learn this magical reward function, but at least the cost of robust optimization might be amortized. In some sense, this result is similar to Ilyas et al. (2019).

## A.10 FINDING THE ROBUST REWARD FUNCTION

In the previous section, we showed that a policy robust against any set of reward functions $R$ can be obtained by solving a MaxEnt RL problem. However, this requires calculating a reward function $r^*$ for MaxEnt RL, which is not in general an element in $R$. In this section, we aim to find the MaxEnt reward function that results in the optimal policy for the robust-reward control problem. Our main idea is to find a reward function $r^*$ such that its robust set, $R_{r^*}$, contains the set of reward functions we want to be robust against, $R$. That is, for each $r_i \in R$, we want

$$r_i(\mathbf{s_t}, \mathbf{a_t}) = r^*(\mathbf{s_t}, \mathbf{a_t}) + u_{\mathbf{s_t}}(\mathbf{a_t}) \quad \text{for some } u_{\mathbf{s_t}}(\mathbf{a_t}) \text{ satisfying} \quad \int_{\mathcal{A}} e^{-u_{\mathbf{s_t}}(\mathbf{a_t})} d\mathbf{a_t} \leq 1 \ \forall \mathbf{s_t}.$$

Replacing $u$ with $r' - r^*$, we see that the MaxEnt reward function $r$ must satisfy the following constraints:

$$\int_{\mathcal{A}} e^{r^*(\mathbf{s_t}, \mathbf{a_t}) - r'(\mathbf{s_t}, \mathbf{a_t})} d\mathbf{a_t} \leq 1 \ \forall \mathbf{s_t} \in \mathcal{S}, r' \in R.$$

We define $R^*(R)$ as the set of reward functions satisfying this constraint w.r.t. reward functions in $R$:

$$R^*(R) \triangleq \left\{ r^* \ \middle| \ \int_{\mathcal{A}} e^{r^*(\mathbf{s_t}, \mathbf{a_t}) - r'(\mathbf{s_t}, \mathbf{a_t})} d\mathbf{a_t} \leq 1 \ \forall \mathbf{s_t} \in \mathcal{S}, r' \in R \right\}$$

Note that we can satisfy the constraint by making $r^*$ arbitrarily negative, so the set $R^*(R)$ is non-empty. We now argue that all any applying MaxEnt RL to any reward function in $r^* \in R^*(R)$ lower bounds the robust-reward control objective.

**Lemma A.4.** *Let a set of reward functions $R$ be given, and let $r^* \in R^*(R)$ be an arbitrary reward function belonging to the feasible set of MaxEnt reward functions. Then*

$$J_{MaxEnt}(\pi, r^*) \leq \min_{r' \in R} \mathbb{E}_{\pi}\left[\sum_{t=1}^{T} r'(\mathbf{s_t}, \mathbf{a_t})\right] \qquad \forall \pi \in \Pi.$$

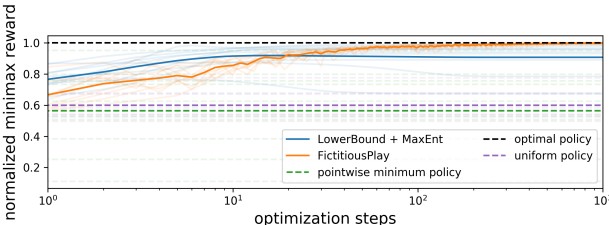

Figure 10: **Approximately solving an arbitrary robust-reward control problem.** In this experiment, we aim to solve the robust-reward control problem for an *arbitrary* set of reward functions. While we know that MaxEnt RL can be used to solve arbitrary robust-reward control problems exactly, doing so requires that we already know the optimal policy (§ A.9). Instead, we use the approach outlined in Sec. A.10, which allows us to *approximately* solve an arbitrary robust-reward control problem without knowing the solution apriori. This approach ("LowerBound + MaxEnt") achieves near-optimal minimax reward.

Note that this bound holds for all feasible reward functions and all policies, so it also holds for the maximum $r^*$:

$$\max_{r^* \in R^*(R)} J_{\text{MaxEnt}}(\pi, r^*) \leq \min_{r' \in R} \mathbb{E}_\pi \left[ \sum_{t=1}^T r'(\mathbf{s_t}, \mathbf{a_t}) \right] \qquad \forall \pi \in \Pi.$$

Defining $\pi^* = \arg\max_\pi J_{\text{MaxEnt}}(\pi, r^*)$, we get the following inequality:

$$\max_{r^* \in R^*(R), \pi \in \Pi} J_{\text{MaxEnt}}(\pi, r^*) \leq \min_{r' \in R} \mathbb{E}_{\pi^*} \left[ \sum_{t=1}^T r'(\mathbf{s_t}, \mathbf{a_t}) \right] \leq \max_{\pi \in \Pi} \min_{r' \in R} \mathbb{E}_\pi \left[ \sum_{t=1}^T r'(\mathbf{s_t}, \mathbf{a_t}) \right]. \quad (20)$$

Thus, we can find the tightest lower bound by finding the policy $\pi$ and feasibly reward $r^*$ that maximize Equation 20:

$$\max_{r, \pi} \quad \mathbb{E}_\pi \left[ \sum_{t=1}^T r(\mathbf{s_t}, \mathbf{a_t}) \right] + \mathcal{H}_\pi[\mathbf{a_t} \mid \mathbf{s_t}] \qquad\qquad (21)$$

$$\text{s.t.} \quad \int_{\mathcal{A}} e^{r(\mathbf{s_t}, \mathbf{a_t}) - r'(\mathbf{s_t}, \mathbf{a_t})} da \leq 1 \; \forall \; \mathbf{s_t} \in \mathcal{S}, r' \in R.$$

It is useful to note that the constraints are simply LOGSUMEXP functions, which are convex. For continuous action spaces, we might approximate the constraint via sampling. Given a particular policy, the optimization problem w.r.t. $r$ has a linear objective and convex constraint, so it can be solved extremely quickly using a convex optimization toolbox. Moreover, note that the problem can be solved independently for every state. The optimization problem is not necessarily convex in $\pi$.

## A.11 ANOTHER COMPUTATIONAL EXPERIMENT

This section presents an experiment to study the approach outlined above. Of particular interest is whether the lower bound (Eq 20) comes close to the optimal minimax policy.

We will solve robust-reward control problems on 5-armed bandits, where the robust set is a collection of 5 reward functions, each is drawn from a zero-mean, unit-variance Gaussian. For each reward function, we add a constant to all of the rewards to make them all positive. Doing so guarantees that the optimal minimax reward is positive. Since different bandit problems have different optimal minimax rewards, we will normalize the minimax reward so the maximum possible value is 1.

Our approach, which we refer to as "LowerBound + MaxEnt", solves the optimization problem in Equation 21 by alternating between (1) solving a convex optimization problem to find the optimal reward function, and (2) computing the optimal MaxEnt RL policy for this reward function. Step 1 is done using CVXPY, while step 2 is done by exponentiating the reward function, and normalizing it to sum to one. Note that this approach is actually solving a harder problem: it is solving the robust-reward control problem for a much larger set of reward functions that contains the original set

of reward functions. Because this approach is solving a more challenging problem, we do not expect that it will achieve the optimal minimax reward. However, we emphasize that this approach may be easier to implement than fictitious play, which we compare against. Different from experiments in Section 4.1, the "LowerBound + MaxEnt" approach assumes access to the full reward function, not just the rewards for the actions taken. For fair comparison, fictitious play will also use a policy player that has access to the reward function. Fictitious play is guaranteed to converge to the optimal minimax policy, so we assume that the minimax reward it converges to is optimal. We compare against two baselines. The "pointwise minimum policy" finds the optimal policy for a new reward function formed by taking the pointwise minimum of all reward functions: $\tilde{r}(\mathbf{a_t}) = \min_{r \in R} r(\mathbf{a_t})$. This strategy is quite simple and intuitive. The other baseline is a "uniform policy" that chooses actions uniformly at random.

We ran each method on the same set of 10 robust-reward control bandit problems. In Fig. 10, we plot the (normalized) minimax reward obtained by each method on each problem, as well as the average performance across all 10 problems. The "LowerBound + MaxEnt" approach converges to a normalized minimax reward of 0.91, close to the optimal value of 1. In contrast, the "pointwise minimum policy" and the "uniform policy" perform poorly, obtaining normalized minimax rewards of 0.56 and 0.60, respectively. In summary, while the method proposed for converting robust-reward control problems to MaxEnt RL problems does not converge to the optimal minimax policy, empirically it performs well.

## B    EXPERIMENTAL DETAILS

### B.1    DYNAMICS ROBUSTNESS EXPERIMENTS (FIG. 4)

We trained the MaxEnt RL policy using the SAC implementation from TF Agents (Guadarrama et al., 2018) with most of the default parameters (unless noted below).

**Fig. 4a**    We used the standard `Pusher-v2` task from OpenAI Gym (Brockman et al., 2016). We used a fixed entropy coefficient of $1e-2$ for the MaxEnt RL results. For the standard RL results, we used the exact same codebase to avoid introducing any confounding factors, simply setting the entropy coefficient to a very small value $1e-5$. The obstacle is a series of three axis-aligned blocks with width 3cm, centered at (0.32, -0.2), (0.35, -0.23), and (0.38, -0.26). We chose these positions to be roughly along the perpendicular bisector of the line between the puck's initial position and the goal. We used 100 episodes of length 100 for evaluating each method. To decrease the variance in the results, we fixed the initial state of each episode:

$$\text{qpos} = [0., 0., 0., 0., 0., 0., 0., -0.3, -0.2, 0., 0.],$$
$$\text{qvel} = [0., 0., 0., 0., 0., 0., 0., 0., 0., 0., 0.].$$

**Fig. 4b**    We used the `SawyerButtonPressEnv` environment from Metaworld (Yu et al., 2020), using a maximum episode length of 151. For this experiment, we perturbed the environment by modifying the observations such that the button appeared to be offset along the $Y$ axis. We recorded the average performance over 10 episodes. For this environment we used an entropy coefficient of $1e1$ for MaxEnt RL and $1e-100$ for standard RL.[2]

**Fig. 6**    We used the standard `Pusher-v2` task from OpenAI Gym (Brockman et al., 2016). We modified the environment to perturb the XY position of the puck at time $\tilde{t} = 20$. We randomly sampled an angle $\theta \sim \text{Unif}[0, 2\pi]$ and displaced the puck in that direction by an amount given by the disturbance size. For evaluating the average reward of each policy on each disturbance size, we used 100 rollouts of length 151.

**Fig. 5**    We used a modified version of the 2D navigation task from Eysenbach et al. (2019) with the following reward function:

$$r(\mathbf{s_t}, \mathbf{a_t}) = \|\mathbf{s_t} - (8, 4)^T\|_2 - 10 \cdot \mathbb{1}(\mathbf{s_t} \in \mathcal{S}_{\text{obstacle}}).$$

Episodes were 48 steps long.

---

[2]We used a larger value for the entropy coefficient for standard RL in the previous experiment to avoid numerical stability problems.

**Fig. 7** We used the peg insertion environment from Eysenbach et al. (2018). Episodes were at most 200 steps long, but terminated as soon as the peg was in the hole. The agent received a reward of +100 once the peg was in the hole, in addition to the reward shaping terms described in Eysenbach et al. (2018).

## B.2 REWARD ROBUSTNESS EXPERIMENTS (FIG. 8)

Following Haarnoja et al. (2018a), we use an entropy coefficient of $\alpha = 1/5$. In Fig. 8, the thick line is the average over five random seeds (thin lines). Fig. 11 shows the evaluation of all methods on both the expected reward and the minimax reward objectives. Note that the minimax reward can be computed analytically as

$$\tilde{r}(\mathbf{s_t}, \mathbf{a_t}) = r(\mathbf{s_t}, \mathbf{a_t}) - \log \pi(\mathbf{a_t} \mid \mathbf{s_t}).$$

## B.3 BANDITS (FIG. 10)

The mean for arm $i$, $\mu_i$, is drawn from a zero-mean, unit-variance Gaussian distribution, $\mu_i \sim \mathcal{N}(0, 1)$. When the agent pulls arm $i$, it observes a noisy reward $r_i \sim \mathcal{N}(\mu_i, 1)$. The thick line is the average over 10 random seeds (thin lines).

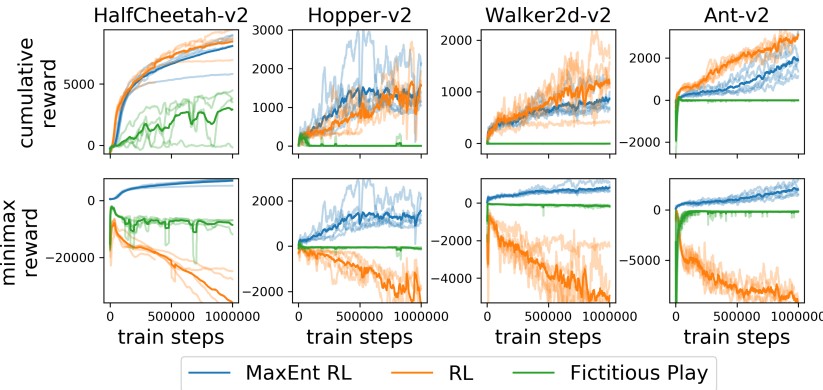

Figure 11: *(Top)* Both RL (SVG) and MaxEnt RL (SAC) effectively maximize expected reward. *(Bottom)* Only MaxEnt RL succeeds in maximizing the minimax reward.

## C ROBUST RL ABLATION EXPERIMENTS

We ran additional ablation experiments to study whether simple modifications to the robust RL baseline from Section 5 (Tessler et al., 2019) would improve results. These experiments will help discern whether the good performance of MaxEnt RL, relative to the PR-MDP and NR-MDP baselines, comes from using a more recent RL algorithm (SAC instead of DDPG), or from the entropy regularization. We used the following ablations:

1. **larger network**: We increased the width of all neural networks to 256 units, so that the network size the exactly the same as for the MaxEnt RL method.

2. **dual critic**: Following TD3 (Fujimoto et al., 2018), we added a second Q function and used the minimum over two (target) Q functions. This change affects not only the actor and critic updates, but also the updates to the adversary learned by PR/NR-MDP.

3. **more exploration**: We increased the exploration noise from 0.2 to 1.0.

We implemented these ablations by modifying the open source code released by Tessler et al. (2019).

The results, shown in Fig. 12, show that these changes do not significantly improve the performance of the NR-MDP or PR-MDP. Perhaps the one exception is on the Hopper-v2 task, where PR-MDP with the larger networks is now the best method for large relative masses. The dual critic ablation generally performs worse than the baseline. We hypothesize that this poor performance is caused by using the (pessimistic) minimum over two Q functions for updating the adversary, which may result

in a weaker adversary. This experiment suggests that incorporating the dual critic trick into the action robustness framework may require some non-trivial design decisions.

(a) NR-MDP ablations.

(b) PR-MDP ablations.

(c) Ablation experiments above overlaid on Fig. 3.

Figure 12: **Robust RL Ablation Experiments**: Ablations of the action robustness framework (Tessler et al., 2019) that use larger networks, multiple Q functions, or perform more exploration do not perform significantly better.

