# OpenReview forum: "Maximum Entropy RL (Provably) Solves Some Robust RL Problems"
_ICLR.cc/2022/Conference — ICLR 2022 Poster_

### Official Review · Reviewer_r4e7 · 2021-10-21

**Correctness:** 4
**Technical Novelty And Significance:** 3
**Empirical Novelty And Significance:** 3
**Recommendation:** 6
**Confidence:** 5

**Main Review:**

Maximum entropy is a form of regularization that prioritizes higher entropy over optimality. Previous works have shown that such regularization may lead to faster learning and in general that there is a connection between generalization <-> regularization <-> robustness.

Hence, while these results are not very surprising the clear connection is important and interesting.

I find the theoretical results and the explanation very good. Maxent will be robust to small pertrubations in the reward/transitions which is similar to what we'd expect in real world robustness where uncertainty often stems from manufacturing defects and sensory-motor degredation.

I do find the empirical results lacking and specifically the focus on NR/PR-MDP (build ontop of DDPG). Action robustness as a framework isn't necessarily limited to the DDPG agent, which raises a question as to whether the performance improvement observed by MaxEnt is due to the learning procedure (larger network capacity, improved stability via dual critics, better exploration via the entropy regularization of maxent) or does the maxent objective itself lead (in these sets of tasks) to improved generalization.

finally, I suggest the authors refer to prior work [1] that analyzes the connection between regularization and robustness in general MDPs (as maximum entropy is a special case of regularization).

[1] Esther Derman, Matthieu Geist, Shie Mannor - Twice regularized MDPs and the equivalence between robustness and regularization


**Summary Of The Paper:**

This work analyzes the implicit robustness of maximum entropy RL. They show, theoretically, that there exists some robust MDP for which the optimal solution is obtained by maximizing the max-ent objective. Empirically they show that indeed SAC learns behavior which is in a sense robust.

**Summary Of The Review:**

interesting paper connecting robustness and maximum entropy.

---

### Official Review · Reviewer_VSYD · 2021-10-27

**Correctness:** 3
**Technical Novelty And Significance:** 4
**Empirical Novelty And Significance:** 2
**Recommendation:** 8
**Confidence:** 3

**Details Of Ethics Concerns:**

No concerns.

**Main Review:**

The paper is generally well written. The motivation of the work is clear, and the results seem technically correct (I did not have time to check the proofs in details, my apologies).

Main comments:

- Some recent references in the field of robust RL with guarantees, are missing:
  1- Reazul Hasan Russel and Marek Petrik. Beyond condence regions: Tight Bayesian ambiguity sets for robust MDPs. Advances in Neural Information Processing Systems, 2019.
  2- Reazul Hasan Russel, Mouhacine Benosman, Jeroen Van Baar, Radu Corcodel, Lyapunov Robust Constrained-MDPs: Soft-Constrained Robustly Stable Policy Optimization under Model Uncertainty, arXiv:2108.02701, 2021.
  3- Reazul Hasan Russel, Bahram Behzadian, Marek Petrik, Entropic Risk Constrained Soft-Robust Policy Optimization, arXiv:2006.11679, 2020.
 whereas other references are not properly cited: Berenkamp et al. 2017 simply uses Lyapunov theory (from control theory) to impose stability of the RL policy, no robustness is analyzed in that work. Similarly for Chow et al. 2018; please either make your statement related to these papers about stability, or remove the references since your work is not about Lyapunov stability of the RL policy.

- Mathematically, the open integrals are rather unnecessary in this context. I assume all your integrations are done over the finite support of action space ? please explain, or amend the paper accordingly.

- The statement about LQG dynamics having same stochasticity at every state, is not clear. Do you mean a dynamical system controlled by an LQG controller ?

- In general the results proposed here in Theorems 4.2 and 4.3 are existence results, i.e., there exists and epsilon, etc. How can these bounds be used to design a policy such that the robustness is guaranteed for a desired epsilon ?

- I found 'reward robustness' examples rather simple. Indeed, with the simple linear dynamics one can compute closed forms of the target reward \bar{r}, etc.but what happens when the dynamics are nonlinear and a closed from solution is not easily obtainable ? do you plan to rely on a numerical search for \bar{r} ?


**Summary Of The Paper:**

The authors work on the important problem of robustness guarantees in RL algorithms. They analyze the robustness of max entropy RL w.r.t. dynamics uncertainties, and reward function uncertainties.

**Summary Of The Review:**

In summary, the paper is a nice step forward in obtaining robustness guarantees for RL algorithms.

---

### Official Review · Reviewer_qrWz · 2021-11-02

**Correctness:** 3
**Technical Novelty And Significance:** 3
**Empirical Novelty And Significance:** 3
**Recommendation:** 6
**Confidence:** 3

**Main Review:**

I find the paper very interesting and in general well-written. Proving that policies learned with entropy regularization are robust in some well-defined sense would be an important result in my opinion. Unfortunately, in the current form, some points are unclear to me (see. The entropy coefficient \alpha in (1) seems to have disappeared in the proofs of Theorems 4.1 and 4.2. Is it related to the KKT multipliers?

Moreover, I believe that the experimental results would have been stronger if the authors had included some experiments with deep RL algorithms based on stochastic policy gradients in contrast to TD3, which yields a deterministic policy. I guess that a stochastic policy would be more robust. How would it fare against the proposed approach?

Detailed comments:
The paper should be checked for typos.
In the appendix, some references to equations or sections are incorrect, e.g.,

-  page 14: Eq. A.2
- Appendix A.3: Section 4.1 -> Section 4.3

In page 13, it would be more rigorous to say that the negative entropy is the Fenchel dual of the log-sum-exp function.

In pages 14 and 16, the authors write "there exists \epislon \ge 0 (the dual solution")…
Could the authors expand this explanation? Why the KKT multipliers don't appear in the relaxed objectives?

In page 16, \Delta(s_{t+1}, s_t, a_t) -> \Delta r(s_{t+1}, s_t, a_t)

In page 17, as time t -> at time t



**Summary Of The Paper:**

The paper shows that maximum entropy RL can yield robust policy with respect to certain disturbances on the reward function and/or transition function.

**Summary Of The Review:**

Well-written paper with interesting results related max entropy RL and robust RL. Some technical results should be clarified and the experimental results could be strengthened.

---

> ### Public Comment · ~Xian_Wu4 · 2022-07-17
> **About the KKT multiplier**
>
> I have the same question as Reviewer qrWz. In pages 14 and 16, the authors write "there exists \epislon \ge 0 (the dual solution")… Could the authors expand this explanation? Why the KKT multipliers don't appear in the relaxed objectives?

---

> > ### Public Comment · ~Benjamin_Eysenbach1 · 2022-07-18
> > **About the KKT multiplier**
> >
> > Thanks for the great question!
> > The Lagrangian would have a term that looks something like
> > $\mathcal{L} = E[\sum \frac{1}{T} \log r(s, a) - \Delta r(s', s, a)] + \lambda \left(E[\sum \log \iint e^{\Delta r(s', s, a')}] - \epsilon \right) + \log T$.
> > By linearity of expectation, this is the same as Eq. 14/15.
> > We use the KKT conditions to argue that there exists a special value of $\epsilon$ for which the constrained objective with that special value of $\epsilon$ has the same solution as the relaxed objective with $\lambda = 1$. Appendix A.6 formally provides a bound on this special value of $\epsilon$.
> >
> > **Does this help explain the result?**

---

> > > ### Public Comment · ~Xian_Wu4 · 2022-07-18
> > > **About the KKT multiplier**
> > >
> > > Thanks for your reply! So Do you mean we could find a $\epsilon$ such that the duality optimal (i.e.,$\lambda^*$ ) is equal to 1 and strong duality holds?

---

> > > > ### Public Comment · ~Benjamin_Eysenbach1 · 2022-07-18
> > > > **About the KKT multiplier**
> > > >
> > > > Yes.

---

### Official Review · Reviewer_bxqc · 2021-11-02

**Correctness:** 2
**Technical Novelty And Significance:** 2
**Empirical Novelty And Significance:** 2
**Recommendation:** 5
**Confidence:** 3

**Main Review:**

The authors investigate an interesting problem, which consists in asking whether maximum entropy reinforcement learning can be thought as a robust problem. The authors correctly describes the related works, which shows the relationship between MaxEnt RL and some kind of robustness w.r.t. the reward.

While the article is indicative of a great effort by authors on the topic, the work still suffers, in my opinion from some major weaknessess. The first one is related the clarity of the paper and the soundness of its claims. After having stated Theorem 4.2, the authors say that it "provides a recipe for converting between MaxEnt RL and robust control problems" and make two claims about that.
1) "On the one hand, if we want to acquire a policy that optimizes a reward under many possible dynamics, we can run MaxEnt RL with a pessimistic reward function". The robust problem that they are lower bounding, though, is a MaxEnt problem too, hence, the obtained policies are not simply optimizing the actual reward under many possible dynamics, since ntropy regularization is involved also in the robust problem.
2) "On the other hand, this result says that every time a user applies MaxEnt RL, they are (implicitly) solving a robust RL problem, one defined in terms of a different reward function.". Also this claim is ambiguous: it is not clear whether the authors are suggesting that the lower bound applies also using any reward in place of $\bar{r}$ and substituting $r$ in the robust problem accordingly, or something else.
Therefore, it is not clear whether the issue is about clarity (in which case the authors should improve their exposition) or whether instead the claims are not sound. It seems that the result presented by the authors connects only two MaxEnt problems, and one of them is a robust one, while the second one has a modified reward. In other words, the theorem gives some insights on the relationship between two regularized problems, but does not seem to relate the regularized problem in any way with some instance of the not-regularized one.

The second issue is related to relevance: to which extent the bound provided by the authors helps in shedding light to MaxEnt optimization? If MaxEnt is related to robust MaxEnt, is it possible to draw a parallel also to standard robustness?

The third weakness is in the experimental analysis, which does not help in clarifying the meaning of Theorem 4.2.
Since results from Theorem 4.1 are known, I would like to suggest the authors to focus more on developing the intuition on Theorem 4.2, removing empirical results supporting the former one.

**Summary Of The Paper:**

The main goal of the paper is to show that optimizing a MaxEnt objective is equivalent to solve a robust problem with uncertainty on the dynamics. The authors provide a theorem which illustrate that solving MaxEnt with a modified reward is equivalent to solve a robust version of MaxEnt w.r.t. an uncertain dynamics. The authors provide some simple examples to show the soundness of their results, and they provide a numerical simulations on more complex problems to experimentally validate their claims.

**Summary Of The Review:**

The authors claim to show a link between robust RL and MaxEnt RL, however, their main result is only about robust MaxEnt RL and MaxEnt RL(with modified reward). The relevance of the paper is, thus, difficult to state, since it is not clear how the result relates with any un-regularized formulation. The clarity of the exposition can also be improved, together with the experimental analysis which struggles to effectively shedding light on the theoretical results. For these reasons I recommend a weak reject.

---

### Decision · Program_Chairs · 2022-01-20

**Decision:**

Accept (Poster)

**Comment:**

The reviewers thought this paper tackles an interesting question around whether MaxEnt RL already provides an important form of robustness. Such work helps us better understand the intersection between generalization, regularization and robustness. The reviewers had a number of comments, questions and clarifications and were generally satisfied with the detailed responses provided by the authors. There was some concern over the strength of the experiments and the authors also ran additional experiments. These addressed one reviewer’s concerns, though the other still thought the existing experiments were a bit too simple.